# Robustness through Random Activation: Adversarial Training with Bernoulli Rectified Linear Units

## Abstract

Despite their considerable achievements across a range of domains, deep learning models have been demonstrated to be susceptible to adversarial attacks. In order to mitigate this vulnerability, adversarial training has become a prevalent defense strategy. In this context, we propose Bernoulli Rectified Linear Units (BReLU), an activation function designed to further enhance the effectiveness of adversarial training. In contrast to conventional activation functions, BReLU modulates activation probabilities in accordance with input values, thereby introducing input-dependent randomness into the model. The experimental results demonstrate that the incorporation of BReLU into adversarial training significantly enhances the robustness of the model against adversarial attacks. Specifically, on the CIFAR-10 dataset using the ResNet-18 model, BReLU improved robustness by 15% under FGSM, by 8% under PGD-20, and by 54% under the CW attack compared to ReLU. Our findings indicate that BReLU represents a promising addition to adversarial training techniques for strengthening deep learning models against adversarial attacks.

## 1 Introduction

Randomness plays a significant role in deep learning models. Weight initialization, dropout, and stochastic gradient descent all involve the random selection of values during the training phase. However, once the model is trained, it becomes deterministic. In other words, if the inputs are the same, the model will always produce the same output. This means that the decision boundary is fixed in the input space, which can make it easier to generate adversarial examples by creating inputs that lie just outside the decision boundary.

White-box adversarial attacks exploit detailed knowledge of the model, such as its architecture and weights, to generate adversarial examples that are both effective and difficult to detect. These attacks aim to find subtle perturbations in the input that push the example just beyond the decision boundary, causing the model to misclassify it. The Fast Gradient Sign Method (FGSM) works by perturbing the input data in the direction of the gradient of the loss function, scaled by a small factor (Goodfellow et al., 2015). Projected Gradient Descent (PGD) improves upon FGSM by applying an iterative approach, refining the perturbations over multiple steps to craft stronger adversarial examples (Madry et al., 2018). At each iteration, the perturbation is projected back into a predefined range to ensure it remains valid. Finally, the Carlini and Wagner (CW) attack diverges from gradient-based approaches by formulating the adversarial example generation as an optimization problem (Carlini & Wagner, 2017). Instead of relying purely on gradients, the CW attack minimizes a combination of two terms—one ensuring the adversarial example closely resembles the original input and another maximizing the attack's success by maximizing the logits of the target class.

To enhance the robustness of deep learning models against adversarial attacks, several defense mechanisms have been proposed. Lecuyer et al. introduced a method called randomized smoothing, which incorporates Gaussian noise between layers to extend the decision boundary of the original models, thereby improving their robustness (Lécuyer et al., 2019; Cohen et al., 2019). Cohen et al. theoretically demonstrated that the addition of noise effectively smooths decision boundaries, increasing the model's resistance to adversarial examples (Cohen et al., 2019). Another adopted de-

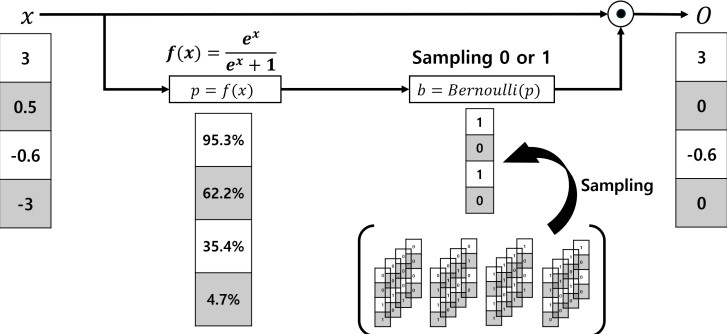

Figure 1: An illustrative toy example of how BReLU operates. First, the probability of a value being 1 is calculated based on the input values. Next, Bernoulli sampling is performed using this calculated probability. Finally, the output is generated by performing element-wise multiplication between the input and the Bernoulli sampling results.

fense strategy is defensive distillation, which reduces the model's sensitivity to small perturbations by training it to replicate the softened output of a pre-trained teacher network (Papernot et al., 2016). In addition, Goodfellow et al. proposed adversarial training, which involves generating adversarial examples immediately after feeding clean data into the model, and using both clean and adversarial examples together during training (Goodfellow et al., 2015). These techniques strengthen the model's resilience to adversarial attacks by broadening the decision boundary.

In this paper, we introduce Bernoulli Rectified Linear Units (BReLU), a novel activation function that integrates Bernoulli sampling with input values to determine activation probabilities, as shown in Figure 1. As the name implies, BReLU is an activation function that can be applied to various types of deep learning models by replacing other activation functions, such as ReLU, ELU, or GELU, with BReLU. In contrast to conventional activation functions, which process inputs in a deterministic manner, BReLU introduces input-dependent randomness into the activation process. We hypothesize that this stochasticity encourages the model to learn more diverse and robust representations, especially under adversarial training conditions, by making the model's behavior less predictable to attackers. To validate our hypothesis, we conducted experiments comparing BReLU with several widely used activation functions and Dropout (Srivastava et al., 2014), evaluating their performance on adversarial robustness across datasets like CIFAR-10 (Krizhevsky & Hinton, 2009) and ImageNet-100 (Russakovsky et al., 2015). Key contributions can be summarized as

- We introduce a novel activation function, BReLU, which adjusts activation probabilities based on input values. We demonstrate the effectiveness of this function in enhancing model robustness when used with adversarial training.

- We empirically show that models trained with BReLU exhibit enhanced robustness against various adversarial attacks, including FGSM, PGD, and CW attacks. Specifically, on the ResNet-18 model with the CIFAR-10 dataset, compared to ReLU, BReLU improved robustness under FGSM by 15%, under PGD-20 by 8%, and under CW by 54%. The significant improvement against the CW attack highlights BReLU's effectiveness in defending against stronger adversarial methods.

- We compare BReLU with Dropout, emphasizing that, in contrast to Dropout, the input-dependent activation mechanism of BReLU effectively enhances model robustness during adversarial training. In the same setting with the ResNet-18 model on CIFAR-10 under PGD-7 attacks, BReLU outperformed Dropout by 16%, demonstrating its superior capability in improving robustness.

- We analyze the role of randomness in input-dependent activation functions. To identify the optimal level of randomness, experiments were conducted, and insights were provided into how stochasticity contributes to improved defense against adversarial attacks.

## 2 BACKGROUND

### 2.1 ADVERSARIAL ATTACK

In this work, we focus on three major white-box attacks: FGSM, PGD, and CW, as they represent a range of adversarial techniques with distinct characteristics. FGSM is computationally efficient and offers a fast method for generating adversarial examples with a single-step approach. PGD, on the other hand, is a multi-step variant that is more robust, offering stronger attacks through iterative updates, and is commonly used as a benchmark in adversarial training. CW, being a more sophisticated attack, minimizes perturbations while bypassing defenses such as adversarial training, making it valuable for evaluating the limits of model robustness under targeted attacks.

#### 2.1.1 FAST GRADIENT SIGN METHOD

FGSM is a relatively simple yet fundamental approach for generating adversarial examples. It perturbs the input data by taking the sign of the gradient of the loss with respect to the input, scaled by a small constant, denoted as $\epsilon$. Adversarial examples are created as $x^* = x + \epsilon \cdot \text{SIGN}(\nabla_x L(\theta, x, y))$, where $x^*$ is the adversarial example, $x$ is the original input data, $\epsilon$ is the magnitude of the perturbation, $L$ is the loss function, $\theta$ represents the model parameters, and $y$ is the label. The perturbation $\epsilon$ is designed to be imperceptible to humans, making the attack subtle yet effective.

#### 2.1.2 PROJECTED GRADIENT DESCENT

PGD is frequently regarded as a more robust variation of FGSM, employing an iterative approach to the attack. The PGD formula can be expressed as $x^t = \prod_{x+S}(x^{t-1} + \alpha \cdot \text{SIGN}(\nabla_x L(\theta, x^{t-1}, y)))$, where $x$ is the original input, $x^t$ is the adversarial example at the $t$-th step, and $\alpha$ represents the step size. The symbol $\prod_{x+S}$ denotes the projection of the perturbation back into the valid input space $S$, ensuring that the adversarial example remains within a certain allowable range. By repeating this process over multiple steps, PGD can generate adversarial examples that are often more effective than those generated by a single-step method like FGSM.

#### 2.1.3 CW ATTACK

CW attack is a white-box attack that optimizes a adversarial example using the model's output logits, differing from gradient-based models like FGSM and PGD. The attack is formulated as an optimization problem, which seeks to create adversarial examples by minimizing two key terms. The optimization objective is to minimize $||\frac{1}{2}(\tanh(w) + 1) - x||_2^2 + c \cdot f(\frac{1}{2}(\tanh(w) + 1))$, where $w$ represents intermediate results used to create an adversarial example. In this equation, the first term ensures that the adversarial example remains similar to the original input $x$ by minimizing the L2 norm between them. The second term controls the success of the attack, where $f(x') = \max(\max\{Z(x')_i : i \neq t\} - Z(x')_t, -\kappa)$. Here, $Z(x')$ represents the logits of the model for input $x'$, and $t$ is the target class. This term ensures that the logits of the target class $t$ become larger than those of any other class by at least $\kappa$, thus increasing the likelihood of the model misclassifying $x'$ as the target class. While the CW attack requires more computational time compared to FGSM and PGD due to its iterative optimization process, it is often more successful in generating adversarial examples that closely resemble the original input while maintaining higher attack efficacy.

### 2.2 ADVERSARIAL TRAINING

Adversarial training (Goodfellow et al., 2015) was introduced as a method to enhance the robustness of models against adversarial attacks. In contrast to the conventional approach of training with only clean samples, adversarial training incorporates adversarial examples generated through adversarial attacks into the training process. The modified loss function utilized in adversarial training can be expressed as follows:

$$\tilde{L}(\theta, x, y) = \alpha L(\theta, x, y) + (1 - \alpha)L(\theta, x^*, y), \tag{1}$$

where $L$ is the loss function, $\theta$ represents the model parameters, $x$ is the input data, $y$ is the corresponding label, $x^*$ is the adversarial example, and $\alpha$ is the coefficient determining the ratio between the the vanilla loss and the loss from the adversarial examples.

In the initial stages of adversarial training research (Goodfellow et al., 2015), FGSM was employed as a single-step attack to rapidly generate adversarial examples. However, models trained with FGSM were found to be susceptible to more sophisticated multi-step attacks, such as PGD. To address this vulnerability, subsequent research (Madry et al., 2018; Lamb et al., 2022) has adopted PGD as the preferred method for generating adversarial examples during adversarial training, resulting in more robust models.

## 3 BERNOULLI RECTIFIED LINEAR UNITS

We propose Bernoulli Rectified Linear Units (BReLU), an activation function that incorporates a Bernoulli sampling based on the input data. While the BReLU methodology can be extended to any activation function, such as ReLU (Nair & Hinton, 2010), Leaky ReLU (LReLU) (Maas et al., 2013), and PReLU (He et al., 2015), this paper focuses on ReLU.

The operation of BReLU involves three steps as shown in Figure 1. The first step is probability calculation. We compute the activation probability using a sigmoid function parameterized by $\alpha$.

$$f(\alpha, x) = \sigma(\alpha, x) = \frac{e^{\alpha x}}{e^{\alpha x} + 1}, \tag{2}$$

where $x$ is the input data, and $\alpha$ controls the slope of the sigmoid curve. In the standard BReLU, we set $\alpha = 1$. The sigmoid function was selected for its advantageous characteristics, namely its capacity to output values between 0 and 1 while effectively mapping input values to probabilities. As the absolute value of the input increases, the sigmoid function converges more sharply toward its limits of 0 or 1, effectively distinguishing between inactive and active states. Additionally, other functions were tested, including periodic functions, but the sigmoid function was found to yield superior performance. This led to the conclusion that the sigmoid function is the optimal choice for probability calculation in BReLU. The second step is Bernoulli sampling. Using the calculated probabilities, we perform Bernoulli sampling to obtain binary values (0 or 1). 0 means deactivation and 1 means activation. Finally, the input of BReLU and the result of Bernoulli sampling are multiplied element-wise.

In BReLU, the activation probability is directly determined by the input value, introducing input-dependent randomness. The sigmoid function ensures probabilities in the range $[0, 1]$, smoothly transitioning between inactive and active states based on the input. The parameter $\alpha$ allows adjustment of the sigmoid curve's steepness. Increasing $\alpha$ sharpens the transition, reducing the range of moderate probabilities (around 0.5), while decreasing $\alpha$ broadens this range. Although we use the sigmoid function for probability calculation, other suitable functions can also be employed depending on the desired activation characteristics.

By applying the reparameterization trick (Kingma & Welling, 2014), BReLU maintains differentiability, ensuring that backpropagation is not disrupted during training. This allows BReLU to be seamlessly integrated into existing models, potentially enhancing their robustness by simply replacing the activation function.

## 4 EXPERIMENTS

### 4.1 BRELU'S PERFORMANCE IN ADVERSARIAL TRAINING

In order to evaluate the effectiveness of BReLU in adversarial training, experiments were conducted to compare it to other activation functions. Three distinct models were utilized for this investigation. The models employed were ResNet-18 (He et al., 2016), VGG-16 (Simonyan & Zisserman, 2015), and EfficientNet-V2 (Tan & Le, 2021). To ensure optimal training on the CIFAR-10 dataset, minor modifications were made to the models (detailed structures are provided in the Appendix). The activation functions in these models were replaced with various alternatives, including ReLU (Nair & Hinton, 2010), Leaky ReLU (LReLU) (Maas et al., 2013), Parametric ReLU (PReLU) (He et al., 2015), GELU (Hendrycks & Gimpel, 2016), SiLU (Ramachandran et al., 2018), and ELU (Clevert et al., 2016).

The models were trained through adversarial training with adversarial examples generated using the PGD-7 attack. The parameters utilized for the PGD attack were set with a perturbation size of

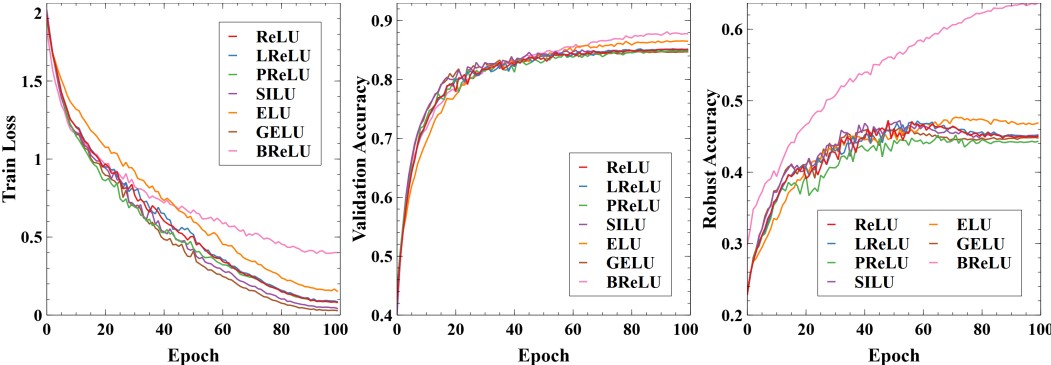

Figure 2: Training loss, validation accuracy, and robust accuracy for the ResNet-18 model on CIFAR-10, comparing seven activation functions. BReLU (shown by the pink curve) exhibits a higher loss across most epochs, indicating slower convergence due to its built-in randomness that prevents overfitting.

$\epsilon = 0.03$, a step size of $\alpha = 0.00784$, and 7 steps. Validation accuracy refers to the accuracy on clean validation data, which is not used for training, while robust accuracy refers to the accuracy on adversarial examples generated from the validation data. In other words, the former reflects performance on clean validation data, while the latter assesses performance on new adversarial examples generated by a PGD-7 attack using the same hyperparameters as during adversarial training. In this context, the PGD attack uses the identical hyperparameters used during adversarial training. The values reported in the tables represent the maximum accuracies achieved over 100 epochs, trained using the AdamW optimizer. Each experiment was repeated five times to calculate the mean and standard deviation.

### 4.1.1 CIFAR-10

Table 1 shows validation and robust accuracy on CIFAR-10 dataset. BReLU exhibits enhanced robust accuracy across all three models. In ResNet-18, BReLU achieves a robust accuracy of 63.9%, which is approximately 16% higher than that achieved by ReLU. This suggests that BReLU is an effective method for enhancing the model's robustness against adversarial examples. In the case of VGG-16, BReLU also demonstrates enhanced robust accuracy, achieving 68.7%, which is approximately 3% higher than the next most effective activation function. ELU also performs well, especially in terms of clean data validation accuracy, but is slightly less robust to adversarial attacks compared to BReLU. In EfficientNet-V2, BReLU enhances robust accuracy by approximately 10% compared to ReLU, although the validation accuracy is slightly lower.

Table 1: Performance comparison of different activation functions in adversarial training on CIFAR-10 (all values are reported in %). BReLU outperforms other functions in terms of robust accuracy in all architectures.

| Activation | ResNet-18 | | VGG-16 | | EfficientNet-V2 | |
| function | Val Acc. | Robust Acc. | Val Acc. | Robust Acc. | Val Acc. | Robust Acc. |
|---|---|---|---|---|---|---|
| ReLU | $85.4 \pm 0.1$ | $47.7 \pm 0.3$ | $85.5 \pm 0.1$ | $63.6 \pm 0.3$ | $\mathbf{81.6 \pm 0.8}$ | $38.9 \pm 1.4$ |
| LReLU | $85.5 \pm 0.1$ | $47.6 \pm 0.2$ | $85.8 \pm 0.2$ | $64.3 \pm 0.3$ | $79.6 \pm 1.4$ | $38.3 \pm 1.4$ |
| PReLU | $85.2 \pm 0.1$ | $46.1 \pm 0.4$ | $87.5 \pm 0.3$ | $65.9 \pm 0.1$ | $76.9 \pm 0.8$ | $33.3 \pm 2.8$ |
| GELU | $84.9 \pm 0.2$ | $47.2 \pm 0.3$ | $86.3 \pm 0.3$ | $64.8 \pm 0.2$ | $81.2 \pm 1.7$ | $40.2 \pm 3.8$ |
| SiLU | $85.5 \pm 0.2$ | $47.6 \pm 0.1$ | $85.3 \pm 0.3$ | $63.7 \pm 0.1$ | $80.3 \pm 2.8$ | $43.1 \pm 1.3$ |
| ELU | $87.0 \pm 0.2$ | $48.1 \pm 0.2$ | $\mathbf{87.6 \pm 0.1}$ | $65.7 \pm 0.2$ | $79.5 \pm 3.7$ | $40.5 \pm 2.1$ |
| BReLU (ours) | $\mathbf{88.2 \pm 0.1}$ | $\mathbf{63.9 \pm 0.2}$ | $85.9 \pm 0.1$ | $\mathbf{68.7 \pm 0.2}$ | $80.1 \pm 1.0$ | $\mathbf{48.9 \pm 1.8}$ |

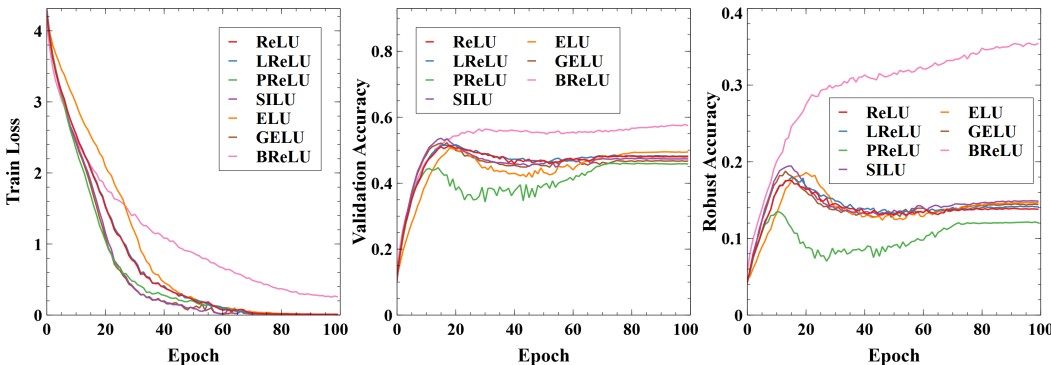

Figure 3: Training loss, validation accuracy, and robust accuracy for the ResNet-18 model on ImageNet-100 dataset, comparing seven activation functions.

### 4.1.2 IMAGENET-100

BReLU was evaluated on the more complex ImageNet-100 dataset, which consists of 100 randomly selected classes from ImageNet-1K (Russakovsky et al., 2015). In these experiments, the standard ResNet-18 model was used without modifications. As demonstrated in Table 2, BReLU markedly outperforms other activation functions on ImageNet-100. BReLU achieves a validation accuracy of 58.1%, approximately 4.2% higher than the next most effective activation function, SiLU. Of greater significance is the fact that BReLU attains a robust accuracy of 35.8%, nearly double that of ReLU (18.2%). This substantial improvement indicates that BReLU effectively enhances robustness in more complex datasets.

### 4.2 PERFORMANCE AGAINST WHITE BOX ATTACKS

To further evaluate the robustness enhancement provided by BReLU, the robustness of the adversarially *trained models* with PGD-7 was evaluated by measuring the classification accuracy on adversarial examples *newly generated* from the validation datasets using three white-box attacks: FGSM, PGD-20, and the CW attack. The maximum perturbation size was set to $\epsilon = 0.0314$ for all attacks.

Models with BReLU demonstrate consistently superior performance in terms of robustness against white-box attacks in comparison to those employing alternative activation functions. The results are summarized in Table 3 for the CIFAR-10 dataset. In ResNet-18, BReLU shows an approximately 15% improvement in accuracy under FGSM compared to ReLU. In response to the stronger PGD-20 attack, BReLU demonstrated a robust accuracy of 50.0%, in comparison to 42.4% with ReLU. Most notably, BReLU exhibits exceptional robustness against the CW attack, achieving an accuracy of 72.4%, which is a significant improvement over the 18.1% achieved by ReLU. In VGG-16, BReLU

Table 2: Adversarial training result on ImageNet-100 data. Val Acc. refers to the accuracy on clean validation data, while Robust Acc. refers to the accuracy on PGD-7 adversarial examples generated from the validation data. The FGSM, PGD-20, and CW columns indicate that the adversarially trained model with PGD-7 is attacked using each respective method. All values are reported as percentages(%).

| Activation function | Val Acc. | Robust Acc. | FGSM | PGD-20 | CW |
|---|---|---|---|---|---|
| ReLU | $52.0 \pm 0.5$ | $18.2 \pm 0.3$ | $20.2 \pm 0.7$ | $15.5 \pm 0.5$ | $43.8 \pm 0.8$ |
| LReLU | $52.4 \pm 0.4$ | $18.3 \pm 0.3$ | $20.6 \pm 0.2$ | $15.4 \pm 0.4$ | $44.4 \pm 0.3$ |
| PReLU | $46.4 \pm 0.5$ | $14.0 \pm 0.6$ | $15.6 \pm 0.6$ | $11.8 \pm 0.4$ | $37.6 \pm 2.1$ |
| GELU | $52.7 \pm 0.5$ | $18.9 \pm 0.3$ | $19.6 \pm 1.0$ | $16.0 \pm 0.6$ | $43.7 \pm 1.8$ |
| SiLU | $53.9 \pm 0.3$ | $19.6 \pm 0.2$ | $21.1 \pm 0.4$ | $17.1 \pm 0.3$ | $46.1 \pm 0.3$ |
| ELU | $51.5 \pm 0.4$ | $18.9 \pm 0.2$ | $20.2 \pm 0.3$ | $16.6 \pm 0.2$ | $44.5 \pm 0.8$ |
| BReLU (ours) | $\mathbf{58.1 \pm 0.2}$ | $\mathbf{35.8 \pm 0.1}$ | $\mathbf{37.7 \pm 1.0}$ | $\mathbf{24.8 \pm 0.4}$ | $\mathbf{55.7 \pm 1.4}$ |

Table 3: Adversarial training result on CIFAR-10 data. The FGSM, PGD-20, and CW columns indicate that the adversarially trained model with PGD-7 is attacked using each respective method. All values are reported as percentages(%).

| Activation function | ResNet-18 | | | VGG-16 | | | EfficientNet-V2 | | |
|---|---|---|---|---|---|---|---|---|---|
| | FGSM | PGD-20 | CW | FGSM | PGD-20 | CW | FGSM | PGD-20 | CW |
| ReLU | $52.3 \pm 0.2$ | $42.4 \pm 0.4$ | $18.1 \pm 2.2$ | $72.4 \pm 0.3$ | $62.2 \pm 0.3$ | $32.5 \pm 0.6$ | $56.6 \pm 3.2$ | $36.6 \pm 1.4$ | $17.3 \pm 1.5$ |
| LReLU | $52.1 \pm 0.4$ | $42.2 \pm 0.4$ | $15.5 \pm 2.6$ | $73.1 \pm 0.4$ | $62.9 \pm 0.4$ | $32.6 \pm 0.3$ | $56.2 \pm 3.8$ | $36.0 \pm 1.1$ | $16.2 \pm 3.6$ |
| PReLU | $51.1 \pm 0.7$ | $40.5 \pm 0.5$ | $18.7 \pm 2.9$ | $\mathbf{75.3 \pm 0.2}$ | $64.4 \pm 0.2$ | $33.5 \pm 0.4$ | $46.7 \pm 4.9$ | $31.5 \pm 2.5$ | $15.2 \pm 1.1$ |
| GELU | $50.7 \pm 0.5$ | $42.4 \pm 0.3$ | $11.4 \pm 2.1$ | $72.8 \pm 0.1$ | $63.8 \pm 0.2$ | $32.7 \pm 0.3$ | $54.1 \pm 6.2$ | $38.8 \pm 3.6$ | $12.9 \pm 4.7$ |
| SILU | $50.8 \pm 0.3$ | $42.7 \pm 0.3$ | $10.9 \pm 1.8$ | $71.7 \pm 0.2$ | $62.5 \pm 0.2$ | $32.0 \pm 0.3$ | $\mathbf{59.0 \pm 1.3}$ | $41.7 \pm 1.4$ | $8.7 \pm 1.0$ |
| ELU | $51.8 \pm 0.2$ | $42.9 \pm 0.5$ | $4.6 \pm 1.1$ | $74.8 \pm 0.3$ | $64.3 \pm 0.3$ | $33.3 \pm 0.4$ | $53.7 \pm 2.5$ | $39.2 \pm 2.1$ | $12.0 \pm 2.4$ |
| BReLU (ours) | $\mathbf{67.0 \pm 0.2}$ | $\mathbf{50.0 \pm 0.6}$ | $\mathbf{72.4 \pm 0.3}$ | $73.1 \pm 0.3$ | $\mathbf{65.0 \pm 0.4}$ | $\mathbf{72.3 \pm 0.6}$ | $51.5 \pm 1.8$ | $\mathbf{46.1 \pm 1.7}$ | $\mathbf{53.7 \pm 1.7}$ |

achieves the highest robust accuracy under PGD-20 (65.0%) and CW attacks (72.3%). Although PReLU achieves the highest accuracy under FGSM (75.3%), BReLU exhibits superior robustness under the more challenging PGD-20 and CW attacks, indicating its effectiveness against stronger adversarial perturbations. In EfficientNet-V2, BReLU enhances robust accuracy under PGD-20, attaining a score of 46.1% in comparison to 36.6% for ReLU. Moreover, BReLU markedly enhances robustness against the CW attack, attaining an accuracy of 53.7%, in comparison to ReLU, which only achieves 17.3%. This substantial improvement underscores BReLU's capacity to reinforce model defenses even in architectures where validation accuracy gains are minimal.

The results for the ImageNet-100 dataset using ResNet-18 are presented in Table 2. The observations indicated that BReLU exhibited a notable enhancement in robustness against white-box attacks. It enhances accuracy by over 17% and 9% respectively, in comparison to ReLU in the context of FGSM and PGD-20 attacks. It is noteworthy that under the CW attack, BReLU achieves an accuracy of 55.7%, in comparison to 43.8% with ReLU. This demonstrates a substantial enhancement in robustness.

It is particularly noteworthy that BReLU demonstrates remarkable performance against the CW attack. Notwithstanding the fact that the models were trained using PGD-based adversarial examples, they exhibited considerable resilience against optimization-based attacks, such as CW. It is postulated that the input-dependent randomness introduced by BReLU results in variability in the logits, thereby rendering it challenging for attacks that rely on precise gradient information to identify optimal adversarial perturbations. This stochastic behavior disrupts the attacker's ability to craft effective adversarial examples, thereby enhancing the model's defense mechanisms.

## 4.3 COMPARISON OF BReLU AND DROPOUT IN ADVERSARIAL TRAINING

In this section, we compare the performance of Dropout (Srivastava et al., 2014) and BReLU to elucidate the differences between these techniques in the context of adversarial training. Although both Dropout and BReLU involve stochastic node disabling, their operational mechanisms are fundamentally distinct. Dropout employs a fixed probability to deactivate nodes only during the training

Table 4: A performance comparison between Dropout and BReLU

| Model | Val Acc. | Robust Acc. |
|---|---|---|
| ReLU | $85.4 \pm 0.1$ | $47.7 \pm 0.3$ |
| ReLU + Dropout ($p = 0.1$) | $86.8 \pm 0.2$ | $46.6 \pm 0.3$ |
| ReLU + Dropout ($p = 0.2$) | $86.6 \pm 0.2$ | $46.7 \pm 0.4$ |
| ReLU + Dropout ($p = 0.5$) | $71.9 \pm 0.5$ | $32.2 \pm 0.2$ |
| LReLU | $85.5 \pm 0.1$ | $47.6 \pm 0.2$ |
| LReLU + Dropout ($p = 0.1$) | $86.9 \pm 0.3$ | $47.2 \pm 0.2$ |
| LReLU + Dropout ($p = 0.2$) | $86.7 \pm 0.2$ | $46.5 \pm 0.2$ |
| LReLU + Dropout ($p = 0.5$) | $72.9 \pm 0.2$ | $32.0 \pm 0.6$ |
| BReLU (ours) | $\mathbf{88.2 \pm 0.1}$ | $\mathbf{63.9 \pm 0.2}$ |

phase, fully activating all nodes during testing and deployment. Conversely, BReLU adjusts the probability of node activation based on the input values, enabling stochastic behavior during both training and testing phases.

The objective of this experiment was twofold: firstly, to ascertain whether Dropout enhances the robustness of models during adversarial training; and secondly, to compare Dropout's performance with that of BReLU. A ResNet-18 model employing ReLU and LReLU activation functions was utilized, with a Dropout layer inserted after the activation functions. The impact of varying dropout probabilities at 10%, 20%, and 50% was examined. The models were trained on CIFAR-10 dataset using adversarial training with a PGD-7 attack, utilizing the AdamW optimizer and hyperparameters outlined in Appendix Table 8.

As illustrated in Table 4, the application of Dropout with a probability of 20% resulted in a marginal enhancement in validation accuracy. However, the robust accuracy decreased for ReLU and showed only a minor improvement for LReLU, with no significant enhancement in performance. As the dropout probability increased, the model's performance consistently declined. These findings indicate that Dropout does not contribute to improving model robustness in adversarial training. In contrast, BReLU demonstrated significantly higher robust accuracy during adversarial training, offering a distinct advantage over Dropout.

### 4.4 Analyzing the Role of Randomness in BReLU

In this section, we investigate the significance of randomness in the BReLU activation function and its impact on model robustness during adversarial training. To understand the effect of randomness, we conducted two key experiments: one involving the replacement of activation functions in an adversarially trained model, and another adjusting the degree of randomness.

#### 4.4.1 Replacing BReLU with Deterministic Activation Functions

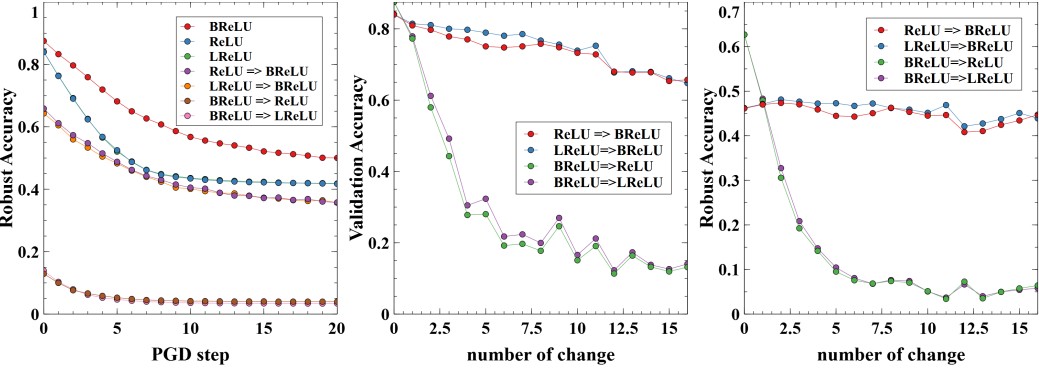

Figure 4: Impact of replacing activation functions on model performance. **Left:** Robust accuracy on adversarial examples versus PGD steps for models with all activation functions replaced. **Middle** and **Right:** Validation and robust accuracy versus the number of activation functions replaced from the input layer, on clean data (Middle) and on adversarial examples generated with a PGD-7 attack (Right). The legends indicate the original and replaced activation functions.

In the first experiment, we investigated the impact of removing randomness from a model that had been trained with BReLU. Specifically, we used adversarial training on a ResNet-18 model with BReLU as the activation function on the CIFAR-10 dataset. Subsequently, the BReLU activation functions were replaced with deterministic alternatives, namely ReLU and LReLU, and the model's performance was evaluated. Conversely, we also replaced the activation functions in a model originally trained with ReLU or LReLU with BReLU to observe the impact of introducing randomness into a deterministically trained model.

The results demonstrated that substituting BReLU with ReLU or LReLU in a model trained with BReLU led to a notable decline in model performance, as evidenced by a substantial reduction in both validation and robust accuracy. This indicates that the randomness intrinsic to BReLU is vital

for the model's functionality, and its removal disrupts the learned representations. Conversely, the introduction of BReLU into a model originally trained with ReLU or LReLU resulted in a notable decline in performance, with the validation accuracy dropping from approximately 84% to around 65%. While there was a decline in overall accuracy, the model exhibited enhanced robustness when subjected to more severe adversarial attacks. Specifically, as the number of PGD steps was increased to intensify the attacks, the model with BReLU exhibited a slower rate of accuracy degradation in comparison to the original model without BReLU (see Figure 4). This indicates that while the introduction of randomness into a deterministically trained model results in a reduction in initial accuracy, it enhances the model's resilience to adversarial perturbations by mitigating the rate at which performance deteriorates under stronger attacks.

These results highlight the essential role of randomness in models trained with BReLU and suggest that the stochastic nature of BReLU contributes significantly to the model's robustness during adversarial training. Removing randomness from a model that relies on it severely degrades performance. Conversely, introducing randomness into a deterministically trained model may reduce initial accuracy but can enhance robustness against stronger adversarial attacks by slowing down the rate of performance degradation.

### 4.4.2 ADJUSTING RANDOMNESS LEVELS WITH VBReLU

In the second experiment, we explored the optimal degree of randomness by using the Variable BReLU (VBReLU) activation function, which introduces an adjustable parameter $\alpha$ to control the level of randomness. By modifying $\alpha$, we can fine-tune the stochastic behavior of the activation function: setting $\alpha < 1$ increases randomness, while $\alpha > 1$ decreases it. When $\alpha = 1$, VBReLU becomes equivalent to BReLU.

We trained ResNet-18 models on CIFAR-10 using adversarial training with varying values of $\alpha$ and recorded the validation and robust accuracies. The results are presented in Table 5 for reference. It was observed that when the value of $\alpha$ was set to a value less than 1, such as 0.05 and 0.1, the models exhibited significantly lower accuracy. An excess of randomness impeded the models' capacity to learn effective representations, as evidenced by the diminished performance observed in the models with $\alpha$ values below 1. At $\alpha = 1$, corresponding to BReLU, the model exhibited the highest performance, indicating an optimal balance of randomness. As the value of $\alpha$ increased beyond 1, the randomness was reduced, and the performance gradually approached that of the ReLU baseline. This trend indicates that a reduction in randomness has the effect of diminishing the benefits provided by BReLU, resulting in performance that is comparable to that of deterministic activation functions.

Table 5: BReLU's performance with different degrees of randomness

| Activation function | $\alpha$ | Val acc. | Robust acc. |
|---|---|---|---|
| ReLU | - | 85.4 | 47.7 |
| VBReLU | 1000 | 85.0 | 47.1 |
| VBReLU | 100 | 85.3 | 47.3 |
| VBReLU | 10 | 85.5 | 48.8 |
| VBReLU | 5 | 86.2 | 51.5 |
| BReLU | 1 | 88.3 | 63.9 |
| VBReLU | 0.5 | 84.0 | 63.4 |
| VBReLU | 0.1 | 60.1 | 47.0 |
| VBReLU | 0.05 | 51.0 | 40.4 |

These experiments indicate that there is an optimal level of randomness that enhances model robustness during adversarial training. Excessive randomness can impair learning by introducing too much stochasticity, while insufficient randomness fails to leverage the benefits of stochastic activation functions like BReLU. Adjusting $\alpha$ allows for fine-tuning of this randomness, and our results suggest that $\alpha = 1$ provides the best balance between randomness and performance.

Table 6: Percentage (%) of negative units before and after activation function

| Layer number | ReLU Pre | ReLU Post | LReLU Pre | LReLU Post | PReLU Pre | PReLU Post | GELU Pre | GELU Post | ELU Pre | ELU Post | SILU Pre | SILU Post | BReLU Pre | BReLU Post |
|---|---|---|---|---|---|---|---|---|---|---|---|---|---|---|
| 1 | 0 | 0 | 45 | 45 | 49 | 49 | 46 | 46 | 70 | 70 | 51 | 51 | 53 | 21 |
| 2 | 0 | 0 | 60 | 60 | 56 | 0 | 56 | 56 | 73 | 73 | 49 | 49 | 41 | 16 |
| 3 | 0 | 0 | 48 | 48 | 41 | 0 | 45 | 45 | 70 | 70 | 51 | 51 | 60 | 20 |
| 4 | 0 | 0 | 70 | 70 | 57 | 0 | 70 | 70 | 78 | 78 | 65 | 65 | 50 | 19 |
| 14 | 0 | 0 | 81 | 81 | 75 | 0 | 79 | 79 | 86 | 86 | 79 | 79 | 56 | 20 |
| 15 | 0 | 0 | 86 | 86 | 81 | 0 | 84 | 84 | 81 | 81 | 85 | 85 | 69 | 20 |
| 16 | 0 | 0 | 91 | 91 | 67 | 0 | 92 | 92 | 91 | 91 | 93 | 93 | 65 | 26 |
| 17 | 0 | 0 | 82 | 82 | 79 | 0 | 78 | 78 | 65 | 65 | 80 | 80 | 64 | 27 |

### 4.4.3 WHY BReLU IS ROBUST AGAINST ADVERSARIAL ATTACK

To investigate why BReLU improves model robustness, we analyzed the proportion of negative units before and after applying various activation functions in a ResNet-18 model using the CIFAR-10 validation set. Table 6 presents the average ratios of negative values, where the layer number indicates the specific layers in the model. For the sake of brevity, we include the entire table into the appendix.

In the case of ReLU, both pre- and post-activation negative ratios are zero, indicating that the network evolves to produce non-negative pre-activation values, effectively making ReLU act as an identity function in those layers. In the case of PReLU, we observed that some layers behave like ReLU with zero negative activations post-activation, suggesting that the learnable parameter adjusts to suppress negative outputs when advantageous. Other activation functions, including LReLU, GELU, ELU, and SiLU maintain negative values after activation. However, the proportion of negative activations tends to increase significantly in deeper layers, at times exceeding 90%. The accumulation of negative values may impede effective learning and gradient propagation.

In contrast, BReLU maintains a balanced and stable ratio of negative activations throughout the network. Before activation, the negative ratio ranges between 40% and 60%; after activation, it consistently remains around 20% across all layers. This stability suggests that BReLU provides a consistent activation distribution, potentially facilitating better learning dynamics. We hypothesize that this balanced maintenance of negative activations contributes to the enhanced robustness observed with BReLU. By preventing the excessive accumulation or suppression of negative activations, BReLU contributes to the maintenance of a healthier activation landscape, thereby rendering the model less susceptible to adversarial perturbations.

Furthermore, the input-dependent randomness introduced by BReLU disrupts the ability of attackers to craft effective adversarial examples by rendering the model's responses less predictable. This effect is particularly significant against optimization-based attacks, such as the CW attack, which relies on precise gradient information. In our experiments, BReLU-trained models exhibited exceptional robustness against CW attacks, with up to a 60% performance improvement over models using standard activation functions. An example illustrating the changes in logits is provided in the Appendix.

## 5 CONCLUSION

In this paper, we propose a novel activation function, BReLU, which integrates Bernoulli sampling with ReLU in order to enhance model robustness. The results of our experiments demonstrate that by introducing BReLU into the model and proceeding with adversarial training, we can markedly enhance the model's robustness through a straightforward replacement of the activation function. Future research will explore the application of BReLU to more complex or smaller datasets and examine its potential in various architectures, including CNN-based and Transformer-based models. Further investigation into optimizing BReLU's stochastic properties may unlock additional performance gains across diverse tasks.

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

# A    MODEL STRUCTURE

## A.1    RESNET-18

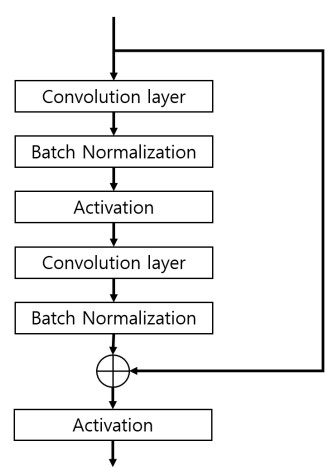

Figure 5: Basic block of ResNet-18

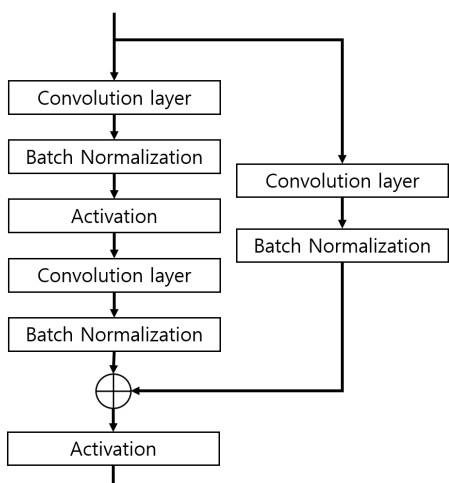

Figure 6: Basic block with downsample of ResNet-18

Table 7: Structure of ResNet-18 for CIFAR-10

| layer name | output size | ResNet-18 |
|---|---|---|
| conv1 | [-1, 64, 32, 32] | Conv 3x3, 64, stride 1
Batch Normalization
Activation function |
| conv2 | [-1, 64, 32, 32] | Basic block x 2 |
| conv3 | [-1, 128, 16, 16] | Basic block with downsample
Basic block |
| conv4 | [-1, 256, 8, 8] | Basic block with downsample
Basic block |
| conv5 | [-1, 512, 4, 4] | Basic block with downsample
Basic block |
| avg | [-1, 512, 1, 1] | average pool |
| fc | [-1, 10] | fully connected |

The default structure is resnet-18 (He et al., 2016) in torchvision.models. Modifications include alterations to the kernel size, stride, and padding in conv1; the absence of a maxpool; and the adaptation of the last fc layer to align with CIFAR-10 standards. The remaining elements remain consistent. When training ImageNet-100, the original structure (He et al., 2016) was utilized without modifications.

## A.2    VGG-16

The base structure is VGG-16 (Simonyan & Zisserman, 2015) from torchvision.models. Modifications include the removal of several maxpools and the resizing of the hidden layer relative to the fully connected layer. Additionally, the dropout, which is added by default, has been removed. The remaining elements are consistent with the original structure.

Table 8: Hyperparameter of ResNet-18

| Hyperparameter | Value |
|---|---|
| optimizer | AdamW |
| batch size | 512 |
| max learning rate | 0.01 |
| weight decay | 0.01 |
| $\beta_1$ | 0.9 |
| $\beta_2$ | 0.999 |
| lr scheduler | OneCycleLR |

Table 9: Structure of VGG-16 for CIFAR-10

| layer name | output size | VGG-16 |
|---|---|---|
| features | [-1, 64, 32, 32] | Conv 3x3, stride 1
Activation function
Conv 3x3, stride 1
Activation function |
| | [-1, 64, 16, 16] | Maxpool 2x2, stride 2 |
| | [-1, 128, 16, 16] | Conv 3x3, stride 1
Activation function
Conv 3x3, stride 1
Activation function |
| | [-1, 128, 8, 8] | Maxpool 2x2, stride 2 |
| | [-1, 256, 8, 8] | Conv 3x3, stride 1
Activation function
Conv 3x3, stride 1
Activation function
Conv 3x3, stride 1
Activation function |
| | [-1, 512, 8, 8] | Conv 3x3, stride 1
Activation function
Conv 3x3, stride 1
Activation function
Conv 3x3, stride 1
Activation function
Conv 3x3, stride 1
Activation function
Conv 3x3, stride 1
Activation function
Conv 3x3, stride 1
Activation function |
| avg | [-1, 512, 7, 7] | Average pool |
| classifier | [-1, 1024] | Fully connected
Activation function
Fully connected
Activation function |
| | [-1, 10] | Fully connected |

## A.3 EFFICIENTNET-V2

The default structure is EfficientNet-V2-S from torchvision.models. Modifications were made to the stride of the first convolution, which was altered from 2 to 1. Additionally, the output of the fully connected layer was modified to ensure compliance with the CIFAR-10 dataset. Finally, the dropout layer was removed, and the remaining elements were left unchanged.

Table 10: Hyperparameter of VGG-16

| Hyperparameter | Value |
|---|---|
| optimizer | AdamW |
| batch size | 512 |
| max learning rate | 0.0005 |
| weight decay | 0.01 |
| $\beta_1$ | 0.9 |
| $\beta_2$ | 0.999 |
| lr scheduler | OneCycleLR |

Table 11: Hyperparameter of EfficientNet-V2-S

| Hyperparameter | Value |
|---|---|
| optimizer | AdamW |
| batch size | 512 |
| max learning rate | 0.05 |
| weight decay | 0.01 |
| $\beta_1$ | 0.9 |
| $\beta_2$ | 0.999 |
| lr scheduler | OneCycleLR |

## B  ANALYZE THE MODEL'S OUTPUT SOFTMAX

Figure 7, 8 demonstrate the effect of different activation functions on the logit values and softmax probbabilities for two images from the CIFAR-10 dataset (ship and frog). These were obtained after adversarial training on a ResNet-18 model using various activation functions: ReLU, LReLU, GELU, and BReLU.

The ship image was correctly classified by all models, as evidenced by the high softmax probabilities across all activation functions. The logit values and confidence scores (softmax outputs) differ slightly across activation functions, but the overall classification remains consistent, indicating that all activations led to robust confidence in the correct class.

BReLU results in multiple outputs due to its inherent randomness. The figures show the highest and lowest confidence scores from five runs. Interestingly, when the confidence is at its peak, BReLU outpurforms other activations, but even in cases where confidence is lower, the model still makes the correct prediction. Other activation functions show high confidence levels, with some variation in the exact confidence percentages, but no change in the predicted class.

For the frog image, the model struggled across all activation functions, resulting in lower softmax probabilities overall. This reflects the challenge in classifying the frog image, as the logits are more spread out across various classes.

GELU failed to classify the image correctly. BReLU, while showing significant variability across runs, consistently classified the image correctly. Despite the fluctuation in logit values and softmax probabilities (with some runs showing very high confidence and others lower confidence), BReLU's randomness seems to have helped the model maintain accuracy. Other activation functions exhibited similar trends, with generally low confidence scores, but they still managed to classify the image correctly in most cases.

One key observation from these figures is that BReLU's logit variations likely contribute to its robustness against CW attacks. The fluctuating logits make it difficult for the CW attack to find a consistent adversarial direction to minimize or maximize specific logits. This randomness in BReLU introduces an additional layer of unpredictability, complicating the adversarial attack's optimization process.

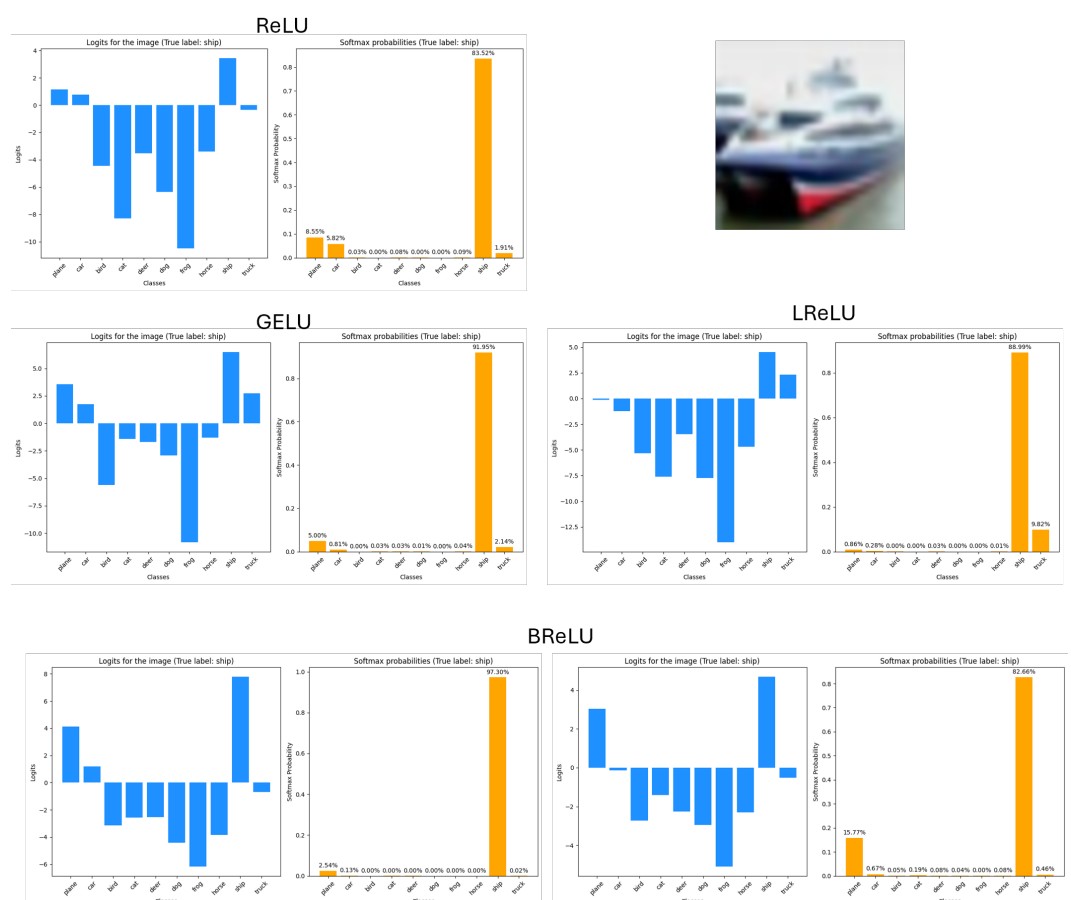

Figure 7: Given a picture of a ship as input, the model's output logit and softmax.

## C  PERCENTAGE OF NEGATIVE NUMBERS BEFORE AND AFTER ACTIVATION FUNCTION

Table 12: Percentage of negative numbers before and after activation function

| Layer number | ReLU Pre | ReLU Post | LReLU Pre | LReLU Post | PReLU Pre | PReLU Post | GELU Pre | GELU Post | ELU Pre | ELU Post | SILU Pre | SILU Post | BReLU Pre | BReLU Post |
|---|---|---|---|---|---|---|---|---|---|---|---|---|---|---|
| 1 | 0 | 0 | 45 | 45 | 49 | 49 | 46 | 46 | 70 | 70 | 51 | 51 | 53 | 21 |
| 2 | 0 | 0 | 60 | 60 | 56 | 0 | 56 | 56 | 73 | 73 | 49 | 49 | 41 | 16 |
| 3 | 0 | 0 | 48 | 48 | 41 | 0 | 45 | 45 | 70 | 70 | 51 | 51 | 60 | 20 |
| 4 | 0 | 0 | 70 | 70 | 57 | 0 | 70 | 70 | 78 | 78 | 65 | 65 | 50 | 19 |
| 5 | 0 | 0 | 42 | 42 | 38 | 0 | 44 | 44 | 70 | 70 | 55 | 55 | 50 | 20 |
| 6 | 0 | 0 | 58 | 58 | 50 | 0 | 60 | 60 | 76 | 76 | 57 | 57 | 42 | 15 |
| 7 | 0 | 0 | 58 | 58 | 55 | 55 | 58 | 58 | 80 | 80 | 62 | 62 | 66 | 17 |
| 8 | 0 | 0 | 75 | 75 | 55 | 0 | 80 | 80 | 88 | 88 | 77 | 77 | 62 | 22 |
| 9 | 0 | 0 | 52 | 52 | 42 | 0 | 53 | 53 | 79 | 79 | 62 | 62 | 53 | 20 |
| 10 | 0 | 0 | 69 | 69 | 59 | 0 | 70 | 70 | 82 | 82 | 70 | 70 | 51 | 18 |
| 11 | 0 | 0 | 70 | 70 | 64 | 0 | 68 | 68 | 89 | 89 | 74 | 74 | 70 | 18 |
| 12 | 0 | 0 | 84 | 84 | 57 | 0 | 87 | 87 | 93 | 93 | 88 | 88 | 64 | 23 |
| 13 | 0 | 0 | 67 | 67 | 45 | 0 | 67 | 67 | 91 | 91 | 77 | 77 | 60 | 22 |
| 14 | 0 | 0 | 81 | 81 | 75 | 0 | 79 | 79 | 86 | 86 | 79 | 79 | 56 | 20 |
| 15 | 0 | 0 | 86 | 86 | 81 | 0 | 84 | 84 | 81 | 81 | 85 | 85 | 69 | 20 |
| 16 | 0 | 0 | 91 | 91 | 67 | 0 | 92 | 92 | 91 | 91 | 93 | 93 | 65 | 26 |
| 17 | 0 | 0 | 82 | 82 | 79 | 0 | 78 | 78 | 65 | 65 | 80 | 80 | 64 | 27 |

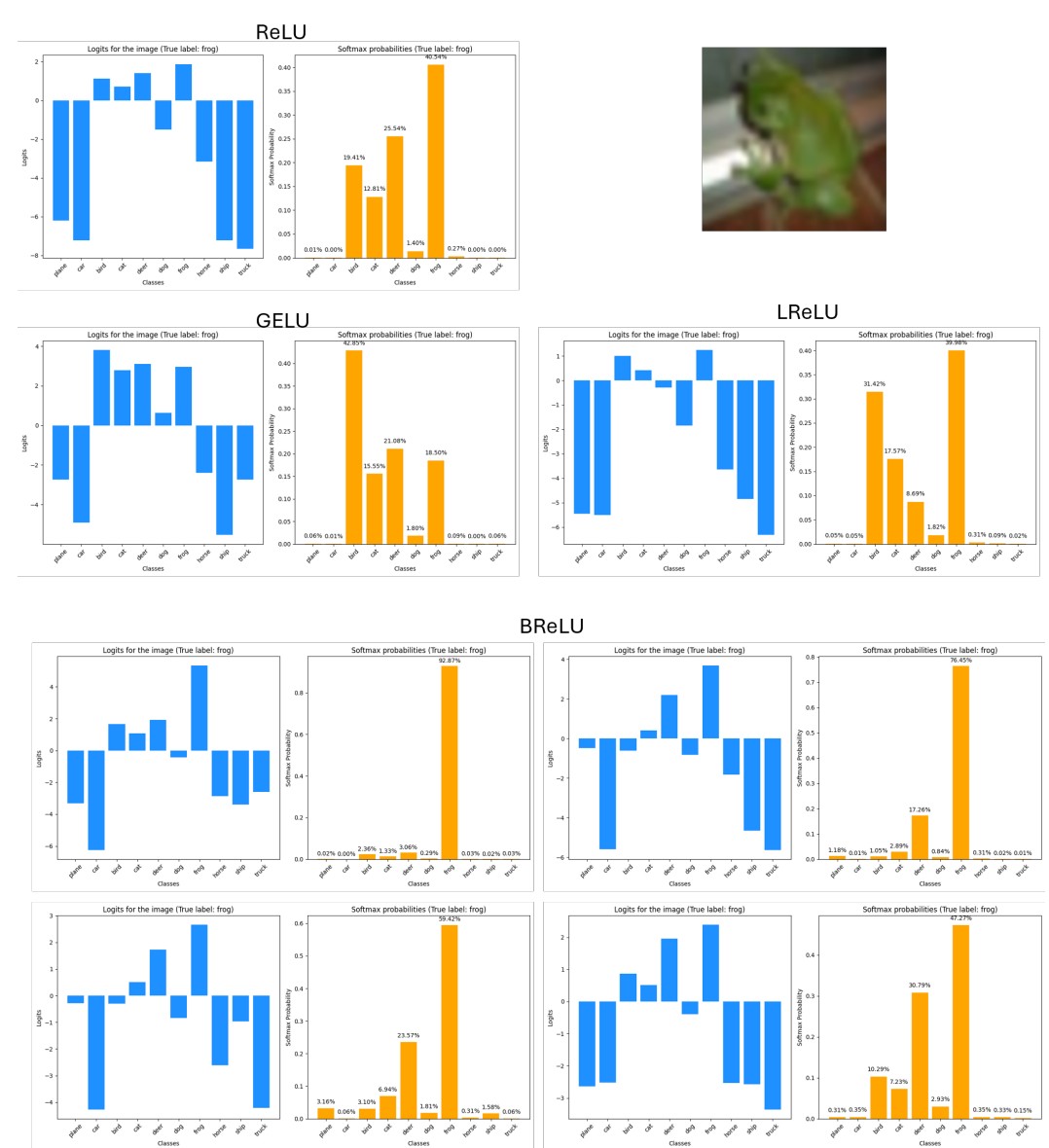

Figure 8: Given a picture of a frog as input, the model's output logit and softmax.

This table is full version of Table 6.

