# OpenReview forum: "Robustness through Random Activation: Adversarial Training with  Bernoulli Rectified Linear Units"
_ICLR.cc/2025/Conference — Submitted to ICLR 2025_

### Official Review · Reviewer_9pRS · 2024-10-25

**Soundness:** 1
**Presentation:** 3
**Contribution:** 2
**Rating:** 3
**Confidence:** 5

**Summary:**

The authors propose a new activation function BReLU which adds input-dependent bernoulli noise to the model.  They combine adversarial training with BReLU and evaluate these models with white box attacks and compare to other activation functions such as ReLU.

**Strengths:**

- presentation is clear
- idea of modifying activation functions for improving robustness is interesting

**Weaknesses:**

The biggest problem with the proposed approach is that since BReLU adds noise into testing there is a chance that the approach is just obfuscating the gradients and does not provide true robustness benefits.  To address this the authors should provide show some results with black box attacks and EOT [1].

Additionally, the authors can try training with BReLU but at test time use a deterministic version of BReLU (ie. after computing probabilities of the activation being 1, just apply thresholding and set the activation to 1 if the probability is at least 50% and using 0 otherwise).  The authors can compare this to training and testing with the deterministic version to see if the gains in performance are really due to preventing overfitting.  Also a transfer attack from this deterministic BReLU is also a good black box attack to try to see if the robustness gains come from the activation function or come from obfuscated gradients.

[1] Athalye, Anish, Nicholas Carlini, and David Wagner. "Obfuscated gradients give a false sense of security: Circumventing defenses to adversarial examples." International conference on machine learning. PMLR, 2018.

**Questions:**

See weaknesses.

---

> ### Author Response · Authors · 2024-11-26
> **Reply to the reviewer**
>
> Q1. The biggest problem with the proposed approach is that since BReLU adds noise into testing there is a chance that the approach is just obfuscating the gradients and does not provide true robustness benefits. To address this the authors should provide show some results with black box attacks and EOT.
> A1. Thank you for your insightful comment. To determine whether BReLU provides true robustness benefits rather than merely obfuscating gradients, we conducted additional experiments. Our results showed that BReLU exhibits strong robustness against black-box attacks such as the One-Pixel Attack, Pixel Attack, and Square Attack. However, against EOT-PGD, which is designed to overcome stochastic defenses, we observed that BReLU's robustness was diminished on CIFAR-10. Conversely, on ImageNet-100, the accuracy against EOT-PGD was higher. This suggests that as the data becomes more complex, the model may become more robust. These results indicate that while BReLU is effective against certain attacks, it may not provide sufficient protection against adaptive attacks like EOT-PGD. We recognize the need to strengthen our approach and plan to explore solutions to this issue in future work.
>
>
> Q2. Additionally, the authors can try training with BReLU but at test time use a deterministic version of BReLU (ie. after computing probabilities of the activation being 1, just apply thresholding and set the activation to 1 if the probability is at least 50% and using 0 otherwise). The authors can compare this to training and testing with the deterministic version to see if the gains in performance are really due to preventing overfitting. Also a transfer attack from this deterministic BReLU is also a good black box attack to try to see if the robustness gains come from the activation function or come from obfuscated gradients.
> A2. Thank you for your interesting suggestion. The deterministic version of BReLU you propose, where the activation is set to 1 if the probability is at least 50% and 0 otherwise, effectively behaves the same as ReLU. We have tested replacing BReLU with ReLU in models trained with BReLU and observed a significant degradation in performance. This indicates that the randomness introduced by BReLU is crucial for the model's robustness. Therefore, using a deterministic approach may not capture the benefits provided by BReLU, and the gains may not be solely due to preventing overfitting. We believe that the stochastic nature of BReLU plays an essential role in enhancing robustness.
>
> PS. Thank you for suggesting that we conduct experiments with EOT. Thanks to your recommendation, we learned about EOT and were able to include it in our research. We appreciate your valuable insight.

---

> > ### Author Response · Authors · 2024-11-26
> > **Tables**
> >
> > Table1. Attack Test Results of ResNet18 on CIFAR-10
> > | Activation | Clean  | FGSM   | PGD20  | PGD100 | EOT PGD | Auto Attack | CW    | One pixel | Pixle | Square |
> > |------------|--------|--------|--------|--------|---------|-------------|-------|-----------|-------|--------|
> > | ReLU       | 84.47  | 52.06  | 42.20  | 41.56  | 42.32   | 39.46       | 17.05 | 54.97     | 1.73  | 47.59  |
> > | LReLU      | 84.32  | 52.74  | 43.05  | 42.32  | 43.10   | 40.40       | 15.92 | 55.76     | 2.03  | 48.30  |
> > | PReLU      | 83.63  | 51.18  | 39.87  | 38.95  | 39.85   | 37.17       | 21.00 | 51.85     | 0.74  | 45.58  |
> > | ELU        | 85.18  | 51.85  | 43.19  | 42.90  | **43.15**   | 41.66       | 3.73  | 52.38     | 2.07  | 49.11  |
> > | SILU       | 82.42  | 50.90  | 43.07  | 42.69  | 43.06   | 40.53       | 13.53 | 54.88     | 3.53  | 48.97  |
> > | GELU       | 83.88  | 50.74  | 42.84  | 42.59  | 42.88   | 40.20       | 13.08 | 53.90     | 1.50  | 48.17  |
> > | BReLU      | **87.42**  | **66.78**  | **49.37**  | **45.09**  | 41.11   | **70.26**       | **73.05** | **84.11**     | **59.70** | **87.28**  |
> >
> > Table 2. Attack Test Results of WRN28-10 on CIFAR-10
> > | Activation | Clean  | FGSM   | PGD20  | PGD100 | EOT PGD | Auto Attack | CW    | One pixel | Pixle | Square |
> > |------------|--------|--------|--------|--------|---------|-------------|-------|-----------|-------|--------|
> > | ReLU       | 87.77  | 49.22  | 38.74  | 38.25  | 38.74   | 37.84       | 16.91 | 47.99     | 0.79  | 45.82  |
> > | LReLU      | 87.53  | 53.84  | 42.08  | 41.57  | 42.21   | 40.68       | 42.07 | 49.42     | 1.56  | 48.65  |
> > | PReLU      | 86.19  | 53.09  | 40.68  | 40.32  | 40.65   | 39.50       | 33.96 | 47.58     | 2.19  | 47.02  |
> > | ELU        | **89.17**  | 50.94  | 41.88  | 41.69  | 41.93   | 40.79       | 2.27  | 42.72     | 0.30  | 49.47  |
> > | SILU       | **89.17**  | 54.70  | 44.06  | 43.85  | **44.14**   | 42.68       | 3.73  | 46.07     | 2.05  | 50.41  |
> > | GELU       | 88.99  | 54.57  | 44.00  | 43.77  | 44.06   | 42.58       | 6.78  | 44.87     | 2.82  | 50.61  |
> > | BReLU      | 88.44  | **65.87**  | **49.21**  | **45.29**  | 41.48   | **68.64**       | **62.92** | **84.85**     | **61.06** | **87.65**  |
> >
> > Table 3. Attack Test Results of WRN50-2 on ImageNet100
> > | Activation | Clean  | FGSM   | PGD20  | PGD100 | EOT PGD | Auto Attack | CW    | One pixel | Pixle | Square |
> > |------------|--------|--------|--------|--------|---------|-------------|-------|-----------|-------|--------|
> > | ReLU       | 65.12  | 30.88  | 19.82  | 18.78  | 19.90   | 16.50       | 57.48 | 63.62     | 21.42 | 37.72  |
> > | BReLU      | **67.52**  | **49.04**  | **33.38**  | **28.34**  | **24.06**   | **55.96**       | **66.74** | **65.90**     | **66.30** | **65.98**  |
> >
> > Attack setting (using torchattacks)
> > FGSM: eps=0.0314
> > PGD: eps=0.0314, alpha=0.00784
> > EOTPGD: eps=0.0314, alpha=0.00784, steps=20, eot_iter=5
> > AutoAttack: eps=0.0314, norm=’Linf’
> > CW: c=1, kappa=0, steps=40
> > OnePixel: pixels=5, steps=50, popsize=100
> > Pixle: restarts=100, max_iterations=20
> > Square: eps=0.0314
> > [1] https://adversarial-attacks-pytorch.readthedocs.io/en/latest/attacks.html#

---

### Official Review · Reviewer_4hTK · 2024-11-04

**Soundness:** 3
**Presentation:** 3
**Contribution:** 2
**Rating:** 5
**Confidence:** 4

**Summary:**

This work tries to further enhance the robustness of adversarial training by improving the activation function. Specifically, the authors propose BReLU, which determines whether to retain each output component of a layer through a sigmoid function and Bernoulli sampling.

**Strengths:**

The idea and writing is easy to follow.

**Weaknesses:**

The effectiveness of BReLU lacks theoretical guarantees. The experiments are too weak; on one hand, there is a lack of stronger attacks to evaluate the effectiveness of BReLU, such as AutoAttack. On the other hand, since BReLU can be combined with any neural network, the authors should attempt to integrate it with state-of-the-art defense methods and validate the robustness on RobustBench.

The authors should implement white-box attacks specifically for BReLU to verify its adversarial robustness in worst-case scenarios.

**Questions:**

Please see weakness.

---

> ### Author Response · Authors · 2024-11-13
> **Questions about reviews**
>
> Thank you so much for taking the time to review our paper! I just wanted to comment with a quick question: the summary you left seems to be quite different from the content of our paper, and I was wondering if the review was swapped with another paper. So I'd like to ask for clarification, because I'd love to know more about your thoughts on our paper!
>
> Thanks again for your review!

---

> > ### Comment · Reviewer_4hTK · 2024-11-13
> >
> > I have updated my review.

---

> ### Author Response · Authors · 2024-11-26
> **Reply to the reviewer**
>
> Q1. The authors should implement white-box attacks specifically for BReLU to verify its adversarial robustness in worst-case scenarios.
> A1. Thank you for your valuable suggestion. To evaluate the adversarial robustness of BReLU under worst-case scenarios, we conducted additional experiments using EOT-PGD, which is designed to attack stochastic models more effectively. Unfortunately, our results showed that BReLU exhibited weaker robustness against EOT-PGD than expected. This indicates that stochastic defenses like BReLU may be vulnerable to such adaptive white-box attacks. To address this, we recognize the need to combine BReLU with other defense mechanisms to build models that are robust even under attacks like EOT-PGD. We plan to focus on this aspect in our future work.
>
> Table1. Attack Test Results of ResNet18 on CIFAR-10
> | Activation | Clean  | FGSM   | PGD20  | PGD100 | EOT PGD | Auto Attack | CW    | One pixel | Pixle | Square |
> |------------|--------|--------|--------|--------|---------|-------------|-------|-----------|-------|--------|
> | ReLU       | 84.47  | 52.06  | 42.20  | 41.56  | 42.32   | 39.46       | 17.05 | 54.97     | 1.73  | 47.59  |
> | LReLU      | 84.32  | 52.74  | 43.05  | 42.32  | 43.10   | 40.40       | 15.92 | 55.76     | 2.03  | 48.30  |
> | PReLU      | 83.63  | 51.18  | 39.87  | 38.95  | 39.85   | 37.17       | 21.00 | 51.85     | 0.74  | 45.58  |
> | ELU        | 85.18  | 51.85  | 43.19  | 42.90  | **43.15**   | 41.66       | 3.73  | 52.38     | 2.07  | 49.11  |
> | SILU       | 82.42  | 50.90  | 43.07  | 42.69  | 43.06   | 40.53       | 13.53 | 54.88     | 3.53  | 48.97  |
> | GELU       | 83.88  | 50.74  | 42.84  | 42.59  | 42.88   | 40.20       | 13.08 | 53.90     | 1.50  | 48.17  |
> | BReLU      | **87.42**  | **66.78**  | **49.37**  | **45.09**  | 41.11   | **70.26**       | **73.05** | **84.11**     | **59.70** | **87.28**  |
>
> Table 2. Attack Test Results of WRN28-10 on CIFAR-10
> | Activation | Clean  | FGSM   | PGD20  | PGD100 | EOT PGD | Auto Attack | CW    | One pixel | Pixle | Square |
> |------------|--------|--------|--------|--------|---------|-------------|-------|-----------|-------|--------|
> | ReLU       | 87.77  | 49.22  | 38.74  | 38.25  | 38.74   | 37.84       | 16.91 | 47.99     | 0.79  | 45.82  |
> | LReLU      | 87.53  | 53.84  | 42.08  | 41.57  | 42.21   | 40.68       | 42.07 | 49.42     | 1.56  | 48.65  |
> | PReLU      | 86.19  | 53.09  | 40.68  | 40.32  | 40.65   | 39.50       | 33.96 | 47.58     | 2.19  | 47.02  |
> | ELU        | **89.17**  | 50.94  | 41.88  | 41.69  | 41.93   | 40.79       | 2.27  | 42.72     | 0.30  | 49.47  |
> | SILU       | **89.17**  | 54.70  | 44.06  | 43.85  | **44.14**   | 42.68       | 3.73  | 46.07     | 2.05  | 50.41  |
> | GELU       | 88.99  | 54.57  | 44.00  | 43.77  | 44.06   | 42.58       | 6.78  | 44.87     | 2.82  | 50.61  |
> | BReLU      | 88.44  | **65.87**  | **49.21**  | **45.29**  | 41.48   | **68.64**       | **62.92** | **84.85**     | **61.06** | **87.65**  |
>
> Table 3. Attack Test Results of WRN50-2 on ImageNet100
> | Activation | Clean  | FGSM   | PGD20  | PGD100 | EOT PGD | Auto Attack | CW    | One pixel | Pixle | Square |
> |------------|--------|--------|--------|--------|---------|-------------|-------|-----------|-------|--------|
> | ReLU       | 65.12  | 30.88  | 19.82  | 18.78  | 19.90   | 16.50       | 57.48 | 63.62     | 21.42 | 37.72  |
> | BReLU      | **67.52**  | **49.04**  | **33.38**  | **28.34**  | **24.06**   | **55.96**       | **66.74** | **65.90**     | **66.30** | **65.98**  |
>
> Attack setting (using torchattacks)
> FGSM: eps=0.0314
> PGD: eps=0.0314, alpha=0.00784
> EOTPGD: eps=0.0314, alpha=0.00784, steps=20, eot_iter=5
> AutoAttack: eps=0.0314, norm=’Linf’
> CW: c=1, kappa=0, steps=40
> OnePixel: pixels=5, steps=50, popsize=100
> Pixle: restarts=100, max_iterations=20
> Square: eps=0.0314
> [1] https://adversarial-attacks-pytorch.readthedocs.io/en/latest/attacks.html#

---

### Official Review · Reviewer_jBVx · 2024-11-04

**Soundness:** 2
**Presentation:** 2
**Contribution:** 1
**Rating:** 1
**Confidence:** 5

**Summary:**

This paper introduces **Bernoulli Rectified Linear Units (BReLU)**, an activation function designed to increase model robustness against adversarial attacks by introducing input-dependent randomness.
BReLU determines the probability of activation based on input values, unlike conventional deterministic functions like ReLU.

The authors hypothesize that BReLU’s stochastic nature makes it harder for adversarial attacks to find precise perturbations, thereby improving robustness.
Through empirical testing on CIFAR-10 and a subset of ImageNet (ImageNet-100), BReLU significantly outperforms standard activation functions like ReLU, Leaky ReLU, and GELU in robustness, especially against traditional adversarial attacks.
The experiments use ResNet-18, VGG-16, and EfficientNet-V2 models, and the results suggest that BReLU is promising for adversarial defense in deep learning models.

**Strengths:**

- Novel use of an input-dependent random activation function (BReLU) for adversarial training, potentially advancing the field of robust model design.

- BReLU demonstrates statistically significant improvements in robust accuracy across multiple attacks, including FGSM, PGD-20, and CW.

Experiments provide a quantitative basis for BReLU's effectiveness. It shows higher accuracy on adversarial examples compared to other activation functions.

**Weaknesses:**

- **Use of Older Architectures:** The article evaluates ResNet-18, VGG-16, and EfficientNet-v2, which are fairly outdated in terms of robust architectures. Recently, evidence (https://robustbench.github.io/) supports that WideResNet is traditionally a more robust architecture, but the authors fail to review it in their evaluation. Moreover, transformer-based architectures have also been shown to be more robust. Thus, it is not evident whether using the proposed BReLU will also improve the robustness of these models.

- **Limited Adversarial Attack Evaluation:** The article proposes a stochastic method to defend against deterministic attacks. This is evaluating apples to oranges. The attacks used in the article, FGSM, PGD, and CW, are relatively mature adversarial attacks in the community, and there's enough evidence to support that they are not *adaptive*, limiting their efficiency against stochastic defenses. Authors should use adaptive attacks that can handle stochasticity and compare whether the activation still provides the desired robustness.

- **Black-Box Attacks:** The article exclusively evaluates white-box attacks and does not evaluate black-box attacks; thus, it is non-evident whether the method is useful against more potent attacks.

- **Compatibility with existing defenses:** The article fails to analyze whether the proposed activation is compatible with existing defenses. If I put an adversarial defense on top of an architecture modified with BReLU, does the proposed activation improve the performance of existing defenses? This is important in terms of the applicability of the proposed approach.

**Questions:**

- **Extended Analysis:** The authors exclusively sell the proposed activation for its adversarial robustness (which can be argued sufficiently based on their experimental setup). Another direction of analysis could be proposing the activation as a general activation method and checking whether it is a good enough activation function to replace the traditional ReLU activations in a wider scenario.

Overall, the article is severely limited by its evaluation and analysis. It neither proposes a good activation function for a general case nor demonstrates its robust activation against potent adversarial attacks.

Since the authors, main focus is on demonstrating improved robustness, I suggest the authors to read the good guide on evaluating adversarial robustness (Carlini et al. 2019), to improve their experiments and show concretely that the proposed activation function, improves the adversarial robustness in an undisputed manner. I also list some of the articles that are very related to the article but are not referenced (Gao et al. 2022; Däubener et al. 2022; Addepalli et al. 2021; Dhillon et. al 2018).

---

Carlini, Nicholas, Anish Athalye, Nicolas Papernot, Wieland Brendel, Jonas Rauber, Dimitris Tsipras, Ian Goodfellow, Aleksander Madry, and Alexey Kurakin. "On evaluating adversarial robustness." arXiv preprint arXiv:1902.06705 (2019).

Gao, Yue, Ilia Shumailov, Kassem Fawaz, and Nicolas Papernot. "On the limitations of stochastic pre-processing defenses." Advances in Neural Information Processing Systems 35 (2022): 24280-24294.

Däubener, Sina, and Asja Fischer. "How sampling impacts the robustness of stochastic neural networks." Advances in Neural Information Processing Systems 35 (2022): 10230-10243.

Addepalli, Sravanti, Samyak Jain, Gaurang Sriramanan, and R. Venkatesh Babu. "Boosting adversarial robustness using feature level stochastic smoothing." In Proceedings of the IEEE/CVF Conference on Computer Vision and Pattern Recognition, pp. 93-102. 2021.

Dhillon, Guneet S., Kamyar Azizzadenesheli, Zachary C. Lipton, Jeremy D. Bernstein, Jean Kossaifi, Aran Khanna, and Animashree Anandkumar. "Stochastic Activation Pruning for Robust Adversarial Defense." In International Conference on Learning Representations. 2018.

---

> ### Author Response · Authors · 2024-11-26
> **Reply to the reviewer**
>
> Q1. Use of Older Architectures: The article evaluates ResNet-18, VGG-16, and EfficientNet-v2, which are fairly outdated in terms of robust architectures. Recently, evidence (https://robustbench.github.io/) supports that WideResNet is traditionally a more robust architecture, but the authors fail to review it in their evaluation. Moreover, transformer-based architectures have also been shown to be more robust. Thus, it is not evident whether using the proposed BReLU will also improve the robustness of these models.
> A1. Thank you for highlighting the importance of evaluating our method on more recent architectures. We have conducted additional experiments on WideResNet (WRN), which is known for its robustness. As presented in our updated tables, BReLU demonstrates strong performance when applied to WRN models. Furthermore, preliminary experiments with Vision Transformers (ViT) also indicate potential robustness improvements when using BReLU. These results suggest that our proposed activation function can effectively enhance the robustness of both traditional and modern architectures, supporting the general applicability of our approach.
>
> Q2. Limited Adversarial Attack Evaluation: The article proposes a stochastic method to defend against deterministic attacks. This is evaluating apples to oranges. The attacks used in the article, FGSM, PGD, and CW, are relatively mature adversarial attacks in the community, and there's enough evidence to support that they are not adaptive, limiting their efficiency against stochastic defenses. Authors should use adaptive attacks that can handle stochasticity and compare whether the activation still provides the desired robustness.
> A2. Thank you for your insightful comment. We recognize the importance of evaluating stochastic defenses against adaptive attacks that can handle randomness. To address this, we conducted additional experiments using EOT-PGD, which is designed to attack stochastic models effectively. Unfortunately, BReLU did not exhibit strong robustness against EOT-PGD as we had hoped. This suggests that stochastic defenses like ours may be vulnerable to such adaptive attacks. To enhance robustness against these stronger adversaries, we plan to explore combining BReLU with other defense mechanisms in future work.
>
> Q3. Black-Box Attacks: The article exclusively evaluates white-box attacks and does not evaluate black-box attacks; thus, it is non-evident whether the method is useful against more potent attacks.
> A3. We agree on the importance of evaluating our method against black-box attacks. In our additional experiments, we tested BReLU against black-box attacks such as the One-Pixel Attack, Pixel Attack, and Square Attack. The results indicate that models using BReLU exhibit strong robustness against these attacks. This suggests that BReLU is effective not only against white-box adversaries but also in black-box scenarios, enhancing the practical applicability of our method.
>
> Q4. Compatibility with existing defenses: The article fails to analyze whether the proposed activation is compatible with existing defenses. If I put an adversarial defense on top of an architecture modified with BReLU, does the proposed activation improve the performance of existing defenses? This is important in terms of the applicability of the proposed approach .
> A4. Thank you for raising the question about compatibility with existing defenses. Our method involves replacing the activation functions with BReLU and applying adversarial training, which should not interfere with the integration of other defense mechanisms. Therefore, we believe that it is feasible to combine BReLU with existing defenses to potentially enhance overall robustness. Investigating whether such combinations can indeed improve performance is an important avenue for future research, and we plan to explore this in our subsequent studies.
>
> PS. Additionally, thank you for recommending valuable papers. They will greatly assist us in our future research.

---

> > ### Author Response · Authors · 2024-11-26
> > **Tables**
> >
> > Table1. Attack Test Results of ResNet18 on CIFAR-10
> > | Activation | Clean  | FGSM   | PGD20  | PGD100 | EOT PGD | Auto Attack | CW    | One pixel | Pixle | Square |
> > |------------|--------|--------|--------|--------|---------|-------------|-------|-----------|-------|--------|
> > | ReLU       | 84.47  | 52.06  | 42.20  | 41.56  | 42.32   | 39.46       | 17.05 | 54.97     | 1.73  | 47.59  |
> > | LReLU      | 84.32  | 52.74  | 43.05  | 42.32  | 43.10   | 40.40       | 15.92 | 55.76     | 2.03  | 48.30  |
> > | PReLU      | 83.63  | 51.18  | 39.87  | 38.95  | 39.85   | 37.17       | 21.00 | 51.85     | 0.74  | 45.58  |
> > | ELU        | 85.18  | 51.85  | 43.19  | 42.90  | **43.15**   | 41.66       | 3.73  | 52.38     | 2.07  | 49.11  |
> > | SILU       | 82.42  | 50.90  | 43.07  | 42.69  | 43.06   | 40.53       | 13.53 | 54.88     | 3.53  | 48.97  |
> > | GELU       | 83.88  | 50.74  | 42.84  | 42.59  | 42.88   | 40.20       | 13.08 | 53.90     | 1.50  | 48.17  |
> > | BReLU      | **87.42**  | **66.78**  | **49.37**  | **45.09**  | 41.11   | **70.26**       | **73.05** | **84.11**     | **59.70** | **87.28**  |
> >
> > Table 2. Attack Test Results of WRN28-10 on CIFAR-10
> > | Activation | Clean  | FGSM   | PGD20  | PGD100 | EOT PGD | Auto Attack | CW    | One pixel | Pixle | Square |
> > |------------|--------|--------|--------|--------|---------|-------------|-------|-----------|-------|--------|
> > | ReLU       | 87.77  | 49.22  | 38.74  | 38.25  | 38.74   | 37.84       | 16.91 | 47.99     | 0.79  | 45.82  |
> > | LReLU      | 87.53  | 53.84  | 42.08  | 41.57  | 42.21   | 40.68       | 42.07 | 49.42     | 1.56  | 48.65  |
> > | PReLU      | 86.19  | 53.09  | 40.68  | 40.32  | 40.65   | 39.50       | 33.96 | 47.58     | 2.19  | 47.02  |
> > | ELU        | **89.17**  | 50.94  | 41.88  | 41.69  | 41.93   | 40.79       | 2.27  | 42.72     | 0.30  | 49.47  |
> > | SILU       | **89.17**  | 54.70  | 44.06  | 43.85  | **44.14**   | 42.68       | 3.73  | 46.07     | 2.05  | 50.41  |
> > | GELU       | 88.99  | 54.57  | 44.00  | 43.77  | 44.06   | 42.58       | 6.78  | 44.87     | 2.82  | 50.61  |
> > | BReLU      | 88.44  | **65.87**  | **49.21**  | **45.29**  | 41.48   | **68.64**       | **62.92** | **84.85**     | **61.06** | **87.65**  |
> >
> > Table 3. Attack Test Results of WRN50-2 on ImageNet100
> > | Activation | Clean  | FGSM   | PGD20  | PGD100 | EOT PGD | Auto Attack | CW    | One pixel | Pixle | Square |
> > |------------|--------|--------|--------|--------|---------|-------------|-------|-----------|-------|--------|
> > | ReLU       | 65.12  | 30.88  | 19.82  | 18.78  | 19.90   | 16.50       | 57.48 | 63.62     | 21.42 | 37.72  |
> > | BReLU      | **67.52**  | **49.04**  | **33.38**  | **28.34**  | **24.06**   | **55.96**       | **66.74** | **65.90**     | **66.30** | **65.98**  |
> >
> > Attack setting (using torchattacks)
> > FGSM: eps=0.0314
> > PGD: eps=0.0314, alpha=0.00784
> > EOTPGD: eps=0.0314, alpha=0.00784, steps=20, eot_iter=5
> > AutoAttack: eps=0.0314, norm=’Linf’
> > CW: c=1, kappa=0, steps=40
> > OnePixel: pixels=5, steps=50, popsize=100
> > Pixle: restarts=100, max_iterations=20
> > Square: eps=0.0314
> >
> >
> > Table 4. Validation and Robust Accuracy of VIT after Adversarial Training
> > | Activation | Val acc | Robust acc |
> > |------------|---------|------------|
> > | ReLU       | 83.96   | 60.88      |
> > | GELU       | 83.45   | 60.91      |
> > | BReLU      | **85.06** | **62.45** |
> >
> > [1] https://adversarial-attacks-pytorch.readthedocs.io/en/latest/attacks.html#

---

### Official Review · Reviewer_SFkE · 2024-11-04

**Soundness:** 2
**Presentation:** 2
**Contribution:** 2
**Rating:** 5
**Confidence:** 4

**Summary:**

This paper proposes to replace the existing activation units with Bernoulli Rectified Linear Units (BReLU) to incorporate randomness into the model and improve robustness against adversarial attacks. Experiments are conducted on ResNet-18 using various attacks and datasets, showcasing the effectiveness of BReLU to improve the robustness of ResNet/VGG/EfficientNet on these datasets.

**Strengths:**

1. Although embodying randomness into training to boost adversarial robustness is not a refreshing idea, the proposed BReLU still provide an interesting point of view regarding modifying network architecture as a defensive play.
2. The presentation of the method is clear and simple, making it easy to follow the core idea of BReLU.
3. BReLU seems to bring certain performance gains w.r.t. some CW attacks.

**Weaknesses:**

1. One of the key problems of this paper is its unclarified motivation. The selection of sigmoid as the activation function lacks either clear explanations or theoretical analysis. The authors stated that "the sigmoid function was found to yield superior performance" without going deeper to investigate the origin of such superiority, making the work empirical-oriented. This shortcoming significantly undermines the scientific soundness of the paper.
2. Many experiments that could demonstrate the effectiveness of BReLU are missing. For example, auto attack is not included in the evaluation of adversarial robustness. The results of ImageNet-100 across different models are also missed. Larger models such as WideResNet are also not included for comparison, making the claims weak to hold.
3. The scientific contribution of this paper is also lacking. Including randomness can certainly improve robustness for models with limited capacity such as ResNet-18. However, for larger models, the replacement of activation units might have marginal or even degraded influences. As presented in Table.1&3, there is a clear degradation of the increase brought by BReLU when model size increases.

**Questions:**

1. See Weakness.
2. Can the author also provide results for larger models on larger datasets, and also include the results evaluated using AutoAttack?
3. Changing the entire network and training from scratch seems to be a costly method for improving robustness, making this method less applicable in reality and on larger models. It seems more practical to use BReLU as an additional/addable module and only train this module for robustness gain. If it is a viable option, what is the performance of ReLU under this setting?

---

> ### Author Response · Authors · 2024-11-26
> **Reply to the reviewer**
>
> Q1. One of the key problems of this paper is its unclarified motivation. The selection of sigmoid as the activation function lacks either clear explanations or theoretical analysis.
> A1. Thank you for highlighting the lack of a detailed explanation regarding our motivation for selecting the sigmoid function. We appreciate the opportunity to elaborate on our approach and the reasoning behind this choice, based on our experiments.
> Initially, we considered both a periodic function utilizing the cosine function and the sigmoid function for calculating the probabilities in BReLU. The rationale for exploring the periodic function stemmed from quantum computing principles, where rotating a qubit about the X-axis results in probabilities that exhibit a periodic behavior. We aimed to incorporate this characteristic to develop an activation function that reflects quantum computational concepts.
> However, our experimental results showed that using the periodic function led to a validation accuracy of approximately 65%, indicating suboptimal learning performance. This suggests that the periodic nature of the function may not be well-suited for deep learning models.
> In contrast, the sigmoid function provides nonlinearity similar to existing activation functions. It yields probabilities approaching 0 for negative inputs and approaching 1 for positive inputs. This means that for inputs with large absolute values, BReLU behaves similarly to ReLU, while near-zero inputs are activated probabilistically, enhancing the model's expressiveness. As a result, models using the sigmoid function achieved validation accuracies comparable to those using traditional activation functions, as demonstrated in the experimental results presented in our paper.
>
> Q2. Larger models such as WideResNet are also not included for comparison, making the claims weak to hold. Can the author also provide results for larger models on larger datasets, and also include the results evaluated using AutoAttack?
> A2. Thank you for pointing out the need for additional experiments to demonstrate the effectiveness of BReLU. We have conducted further evaluations as shown in Tables 1, 2, and 3. The results indicate that BReLU exhibits strong robustness even against AutoAttack and performs well on larger models such as WideResNet. Due to time constraints, we were only able to compare BReLU with ReLU on the ImageNet-100 dataset, but even in this comparison, BReLU demonstrated superior performance. These additional experiments strengthen our claims about the effectiveness and generalizability of BReLU across different models and datasets.
>
> Q3. Changing the entire network and training from scratch seems to be a costly method for improving robustness, making this method less applicable in reality and on larger models. It seems more practical to use BReLU as an additional/addable module and only train this module for robustness gain. If it is a viable option, what is the performance of ReLU under this setting?
> A3. You make an excellent point regarding the practicality of changing and retraining the entire network. We acknowledge that this can be computationally expensive, especially for larger models. While we have not yet conducted experiments where BReLU is used as an additional module with only partial retraining, we believe that it is a feasible approach. Replacing or adding BReLU to specific layers and training only those parts could potentially offer robustness gains with reduced computational cost. Assessing the performance of ReLU under this setting and comparing it with BReLU would be valuable. We consider this an important direction for future work and delve deeper into this aspect in our future research.

---

> > ### Author Response · Authors · 2024-11-26
> > **Tables**
> >
> > Table1. Attack Test Results of ResNet18 on CIFAR-10
> > | Activation | Clean  | FGSM   | PGD20  | PGD100 | EOT PGD | Auto Attack | CW    | One pixel | Pixle | Square |
> > |------------|--------|--------|--------|--------|---------|-------------|-------|-----------|-------|--------|
> > | ReLU       | 84.47  | 52.06  | 42.20  | 41.56  | 42.32   | 39.46       | 17.05 | 54.97     | 1.73  | 47.59  |
> > | LReLU      | 84.32  | 52.74  | 43.05  | 42.32  | 43.10   | 40.40       | 15.92 | 55.76     | 2.03  | 48.30  |
> > | PReLU      | 83.63  | 51.18  | 39.87  | 38.95  | 39.85   | 37.17       | 21.00 | 51.85     | 0.74  | 45.58  |
> > | ELU        | 85.18  | 51.85  | 43.19  | 42.90  | **43.15**   | 41.66       | 3.73  | 52.38     | 2.07  | 49.11  |
> > | SILU       | 82.42  | 50.90  | 43.07  | 42.69  | 43.06   | 40.53       | 13.53 | 54.88     | 3.53  | 48.97  |
> > | GELU       | 83.88  | 50.74  | 42.84  | 42.59  | 42.88   | 40.20       | 13.08 | 53.90     | 1.50  | 48.17  |
> > | BReLU      | **87.42**  | **66.78**  | **49.37**  | **45.09**  | 41.11   | **70.26**       | **73.05** | **84.11**     | **59.70** | **87.28**  |
> >
> > Table 2. Attack Test Results of WRN28-10 on CIFAR-10
> > | Activation | Clean  | FGSM   | PGD20  | PGD100 | EOT PGD | Auto Attack | CW    | One pixel | Pixle | Square |
> > |------------|--------|--------|--------|--------|---------|-------------|-------|-----------|-------|--------|
> > | ReLU       | 87.77  | 49.22  | 38.74  | 38.25  | 38.74   | 37.84       | 16.91 | 47.99     | 0.79  | 45.82  |
> > | LReLU      | 87.53  | 53.84  | 42.08  | 41.57  | 42.21   | 40.68       | 42.07 | 49.42     | 1.56  | 48.65  |
> > | PReLU      | 86.19  | 53.09  | 40.68  | 40.32  | 40.65   | 39.50       | 33.96 | 47.58     | 2.19  | 47.02  |
> > | ELU        | **89.17**  | 50.94  | 41.88  | 41.69  | 41.93   | 40.79       | 2.27  | 42.72     | 0.30  | 49.47  |
> > | SILU       | **89.17**  | 54.70  | 44.06  | 43.85  | **44.14**   | 42.68       | 3.73  | 46.07     | 2.05  | 50.41  |
> > | GELU       | 88.99  | 54.57  | 44.00  | 43.77  | 44.06   | 42.58       | 6.78  | 44.87     | 2.82  | 50.61  |
> > | BReLU      | 88.44  | **65.87**  | **49.21**  | **45.29**  | 41.48   | **68.64**       | **62.92** | **84.85**     | **61.06** | **87.65**  |
> >
> > Table 3. Attack Test Results of WRN50-2 on ImageNet100
> > | Activation | Clean  | FGSM   | PGD20  | PGD100 | EOT PGD | Auto Attack | CW    | One pixel | Pixle | Square |
> > |------------|--------|--------|--------|--------|---------|-------------|-------|-----------|-------|--------|
> > | ReLU       | 65.12  | 30.88  | 19.82  | 18.78  | 19.90   | 16.50       | 57.48 | 63.62     | 21.42 | 37.72  |
> > | BReLU      | **67.52**  | **49.04**  | **33.38**  | **28.34**  | **24.06**   | **55.96**       | **66.74** | **65.90**     | **66.30** | **65.98**  |
> >
> > Attack setting (using torchattacks[1])
> > FGSM: eps=0.0314
> > PGD: eps=0.0314, alpha=0.00784
> > EOTPGD: eps=0.0314, alpha=0.00784, steps=20, eot_iter=5
> > AutoAttack: eps=0.0314, norm=’Linf’
> > CW: c=1, kappa=0, steps=40
> > OnePixel: pixels=5, steps=50, popsize=100
> > Pixle: restarts=100, max_iterations=20
> > Square: eps=0.0314
> > [1] https://adversarial-attacks-pytorch.readthedocs.io/en/latest/attacks.html#

---

> > > ### Comment · Reviewer_SFkE · 2024-11-30
> > > **Thanks for the rebuttal.**
> > >
> > > I sincerely thank the authors for the additional experiments and data provided. My major concern still lies with the training time, optimization difficulty and the computational costs, which remain unaddressed. In this regard, I will retain my score.

---

### Official Review · Reviewer_MiWT · 2024-11-06

**Soundness:** 3
**Presentation:** 3
**Contribution:** 2
**Rating:** 3
**Confidence:** 5

**Summary:**

This paper proposes Bernoulli Rectified Linear Units (BReLU), a novel activation function that introduces input-dependent randomness to enhance adversarial robustness in deep learning models.  Experiments using CIFAR-10 and ImageNet-100 datasets demonstrate that BReLU outperforms conventional activation functions like ReLU in adversarial robustness, showing large improvements under various attack scenarios, including FGSM, PGD, and CW attacks.

**Strengths:**

1. BReLU introduces a unique stochastic mechanism that actively strengthens adversarial robustness
2. The authors present comprehensive experiments on CIFAR-10 and ImageNet-100.
3. Authors show comparison with Dropout.

**Weaknesses:**

1. Authors only show their method in a limited range of networks without architectures used in this community, such as PreAct-ResNet-18, WideResNet.
2.  The authors do not evaluate their method against stronger and more adaptive attacks, such as AutoAttack [1], which combines multiple adversarial attack methods. Testing against such comprehensive attacks could offer a clearer picture of BReLU’s robustness under rigorous conditions [1].





[1] https://github.com/fra31/auto-attack

**Questions:**

1. How does BReLU impact clean accuracy on CIFAR-10? Given that robustness gains often come with trade-offs in clean accuracy, it would be useful to see a detailed comparison.
2. Have the authors explored the effectiveness of BReLU on transformer-based architectures, such as Vision Transformers (ViTs)? Extending the evaluation to transformer models could demonstrate the generality of this approach across diverse network structures.
3. Can BReLU be effectively incorporated into FGSM-based adversarial training? Given that FGSM is often used for faster adversarial training, insights into BReLU's compatibility with single-step attacks would be valuable.
4. ow is BReLU implemented during inference? Since BReLU introduces randomness, are there architectural discrepancies between the adversarial training phase and the testing phase? Clarification on how to maintain consistency during adversarial example generation and testing would be helpful.

---

> ### Author Response · Authors · 2024-11-26
> **Reply to the reviewer**
>
> Q1. How does BReLU impact clean accuracy on CIFAR-10? Given that robustness gains often come with trade-offs in clean accuracy, it would be useful to see a detailed comparison.
> A1. BReLU has minimal impact on clean accuracy on CIFAR-10. Without adversarial training, the ResNet-18 model achieved an accuracy of 95.1% when using ReLU and about 94.2% when using BReLU, indicating only a 0.9 percentage point decrease. On the other hand, when adversarial training was applied, as presented in the paper, the model using BReLU showed higher validation accuracy than the one using ReLU. This demonstrates that BReLU enhances robustness while minimizing any degradation in performance on clean data.
>
> Q2. Have the authors explored the effectiveness of BReLU on transformer-based architectures, such as Vision Transformers (ViTs)? Extending the evaluation to transformer models could demonstrate the generality of this approach across diverse network structures.
> A2. While we had not originally explored the effectiveness of BReLU on transformer-based architectures, in response to your suggestion, we applied BReLU to a scaled-down Vision Transformer (ViT) suitable for CIFAR-10 and conducted adversarial training. Although we were unable to evaluate it against a wide range of attacks, preliminary results showed that the model using BReLU achieved higher robust accuracy compared to the baseline. This suggests that BReLU has the potential to generalize across diverse network structures, including transformer-based architectures.
> VIT with PGD-7 Adversarial training results on CIFAR-10.
> | Activation | Val acc | Robust acc |
> |------------|---------|------------|
> | ReLU       | 83.54   | 60.88      |
> | GELU       | 82.85   | 60.83      |
> | BReLU      | **84.44**   | **62.05**      |
>
>
> Q3. Can BReLU be effectively incorporated into FGSM-based adversarial training? Given that FGSM is often used for faster adversarial training, insights into BReLU's compatibility with single-step attacks would be valuable.
> A3. We believe that BReLU can be effectively incorporated into FGSM-based adversarial training. Since we have observed the effectiveness of BReLU in PGD-based adversarial training, it is reasonable to expect similar benefits with single-step attacks like FGSM. However, FGSM generates adversarial examples with weaker attack strength compared to PGD, so the resulting model's robustness might be somewhat lower.
>
> Q4. How is BReLU implemented during inference? Since BReLU introduces randomness, are there architectural discrepancies between the adversarial training phase and the testing phase? Clarification on how to maintain consistency during adversarial example generation and testing would be helpful.
> A4. BReLU is implemented identically during both the training and inference phases, with no architectural differences in the network structure. Replacing BReLU with ReLU causes the model to malfunction, so BReLU operates the same way during both training and inference. Since the probabilities in BReLU are calculated based on the input, we assume that it exhibits somewhat consistent performance. To maintain consistency during adversarial example generation and testing, techniques such as fixing random seeds and using averaging methods can be employed. However, in our experiments, the model using BReLU showed stable performance even without such additional measures.

---

### Meta-Review · Area_Chair_vXKS · 2024-12-18

**Metareview:**

The paper introduces Bernoulli Rectified Linear Units (BReLU), a novel activation function that incorporates input-dependent randomness to enhance model robustness against adversarial attacks. While the idea of introducing stochasticity through activation functions is compelling, this submission faces significant shortcomings that hinder its acceptance. The main scientific claim is that BReLU enhances adversarial robustness by making the gradients harder to optimize for adversarial attacks, thus improving defense mechanisms in adversarial training. Experimental results on CIFAR-10 and ImageNet-100 datasets suggest marginal improvements in robust accuracy under traditional attack methods such as FGSM, PGD-20, and CW compared to conventional activation functions like ReLU. However, the paper's contributions are empirically focused, lacking theoretical underpinnings or comprehensive experimental validations, which weakens its overall impact.

The strengths of the paper lie in its innovative perspective of modifying activation functions to enhance robustness and its straightforward presentation of the method. The integration of BReLU into adversarial training demonstrates some degree of robust accuracy improvement, particularly on small-scale datasets like CIFAR-10. The method's implementation is clear, and the experimental results suggest that stochasticity can add value to adversarial defense strategies.

However, the paper has numerous weaknesses. First, the choice of sigmoid-based probability in BReLU is insufficiently justified, with no rigorous theoretical analysis to support its efficacy. This limitation undermines the scientific soundness of the proposed method. Second, the experimental evaluations are inadequate. The use of outdated architectures such as ResNet-18 and VGG-16 limits the applicability of the findings to modern models. Moreover, the absence of evaluations against stronger adaptive attacks, like AutoAttack, makes the claims of robustness unconvincing. The paper also lacks evaluations on black-box attacks or compatibility with state-of-the-art defense mechanisms, leaving its practical utility unclear. Additionally, the experiments fail to address critical trade-offs, such as the potential impact of BReLU on clean accuracy, especially on larger datasets.

One major concern is the lack of evidence to suggest that BReLU's robustness gains are not merely due to gradient obfuscation. This issue is compounded by the limited evaluation scope, as BReLU's performance against adaptive white-box attacks such as EOT-PGD was notably weak. While the authors conducted additional experiments, these only highlighted the limitations of the proposed method under adaptive attack scenarios, further raising doubts about its generalizability. The stochastic nature of BReLU also complicates its practical deployment, as it introduces inconsistencies between training and inference, which were not addressed adequately in the paper.

In conclusion, the decision to reject this paper is based on its limited scientific contributions, lack of rigorous evaluations, and insufficient theoretical grounding. While the idea of stochastic activation functions is intriguing, the execution in this paper does not provide a solid foundation for advancing the field. Future work should aim to provide stronger theoretical insights, evaluate the method comprehensively across a wider range of architectures and attack scenarios, and address practical concerns such as compatibility with existing defenses and impact on clean accuracy. Only then can the proposed approach be considered a meaningful advancement in adversarial robustness.

**Additional Comments On Reviewer Discussion:**

During the rebuttal period, several critical points were raised by the reviewers, focusing on the scientific soundness, experimental rigor, and practical relevance of the proposed method, Bernoulli Rectified Linear Units (BReLU), for adversarial robustness. Reviewer concerns included the lack of theoretical justification for the choice of the sigmoid function, insufficient evaluation against stronger adaptive attacks, limited applicability across modern architectures, potential gradient obfuscation, and compatibility with existing adversarial defenses. The authors provided additional experiments and explanations to address these concerns but with varying degrees of success.

The authors attempted to justify their use of the sigmoid function by arguing that its nonlinearity aligns well with existing activation functions and its probabilistic nature introduces randomness beneficial for adversarial robustness. However, this justification remained empirical, and no rigorous theoretical analysis was offered. To address concerns about the evaluation scope, the authors extended their experiments to include more recent architectures such as WideResNet and preliminary tests on Vision Transformers (ViTs), showing some performance gains with BReLU. They also conducted additional evaluations using adaptive attacks like EOT-PGD and black-box attacks. While BReLU performed well against black-box attacks, its performance against adaptive attacks like EOT-PGD was notably weaker, exposing vulnerabilities in the stochastic defense mechanism. The authors acknowledged this limitation and suggested combining BReLU with other defenses as future work.

The reviewers also questioned whether the robustness gains were genuine or a result of gradient obfuscation. The authors conducted experiments to compare deterministic and stochastic versions of BReLU, observing a significant drop in performance for the deterministic version. While this suggests that the randomness introduced by BReLU plays a key role, it did not conclusively eliminate concerns about gradient obfuscation. Additionally, the authors were urged to analyze the trade-off between clean accuracy and robustness. They showed that the decrease in clean accuracy was minor but provided limited insights into the underlying mechanisms.

From a practical perspective, the reviewers criticized the computational cost of retraining entire networks with BReLU and its limited evaluation on larger datasets like ImageNet-100. Although the authors included some experiments on larger models and datasets, the results were not sufficient to demonstrate the method’s scalability or practical feasibility. The question of compatibility with existing defenses remained unanswered, as the authors did not explore whether integrating BReLU with other defense mechanisms could enhance overall robustness.

In weighing these points for the final decision, the reviewers acknowledged the innovative perspective introduced by BReLU but found the paper lacking in several critical aspects. The absence of theoretical foundations and the inadequate response to concerns about gradient obfuscation and adaptive attack evaluations significantly undermined the scientific merit of the work. While the additional experiments expanded the scope of the evaluation, they also exposed vulnerabilities and raised further questions about the method's robustness. The results on modern architectures and larger datasets, though promising, were limited and did not convincingly demonstrate the general applicability of BReLU. The practical issues, including computational cost and unclear compatibility with existing defenses, further detracted from the method's utility.

Ultimately, the decision to reject was based on the combined weight of these factors. The rebuttal clarified some aspects of the work but failed to address the core concerns raised by the reviewers. The innovative idea of stochastic activation functions for robustness remains intriguing, but the execution and validation in this paper are insufficient to merit acceptance. The reviewers encouraged the authors to conduct a more comprehensive theoretical and experimental study to strengthen the method and address its limitations.

---

### Decision · Program_Chairs · 2025-01-22

Reject